# Multiple Early Life Stressors as Risk Factors for Neurodevelopmental Abnormalities in the F1 Wistar Rats

**DOI:** 10.3390/brainsci13101360

**Published:** 2023-09-22

**Authors:** Syed Mujtaba, Ishan Kumar Patro, Nisha Patro

**Affiliations:** 1School of Studies in Neuroscience, Jiwaji University, Gwalior 474011, India; mujtabasyed.ju@gmail.com (S.M.); ikpatro@jiwaji.edu (I.K.P.); 2School of Studies in Zoology, Jiwaji University, Gwalior 474011, India

**Keywords:** multi-hit stress, protein malnourishment, low anxiety, hyperactivity, schizophrenia, ADHD

## Abstract

Cumulative exposure to multiple early life stressors is expected to affect behavioral development, causing increased susceptibility to neuropsychiatric disorders. The present study was designed to mimic such conditions in a rat model to study behavioral impairments during adolescence and adulthood. Female Wistar rats (n = 32; 140–150 gm) were switched to a low protein (LP; 8% protein) or control (20% protein) diet 15 days prior to conception, and then the diet regime was maintained throughout the experimental period. Pups born to control and LP dams were intraperitoneally injected with deltamethrin (DLT—pyrethroid insecticide; 0.7 mg/kg body weight; PND 1 to 7), lipopolysaccharide (LPS—bacterial endotoxin; 0.3 mg/kg body weight; PND 3 and 5), or DLT+LPS, on designated days forming eight experimental groups (Control, LP, Control+LPS, LP+LPS, Control+DLT, LP+DLT, Control+DLT+LPS and LP+DLT+LPS). Neurobehavioral assessments were performed in F1 rats (1, 3, 6 months) by open field, elevated plus maze, light and dark box, and rotarod tests. LP rats were found to be highly susceptible to either singular or cumulative exposure as compared to their age-matched control counterparts, showing significantly severe behavioral abnormalities, such as hyperactivity, attention deficits and low anxiety, the hallmark symptoms of neuropsychiatric disorders like schizophrenia and ADHD, suggesting thereby that early life multi-hit exposure may predispose individuals to developmental disorders.

## 1. Introduction

The developing brain is highly susceptible to adverse environmental conditions, leading to many potential and permanent structural alterations and adverse long-term effects on the behavioral and cognitive abilities of an individual. The early stages of life are marked by rapid brain growth and a dynamic process of synapse sculpting and pruning, making this time particularly prone to harmful disruptions [1]. Perinatal stressors like protein malnutrition, infections and neurotoxicant exposure have been implicated as risk factors for developing brain, leading to cognitive, behavioral and emotional impairments in both animals and humans, raising susceptibility to neuropsychiatric and neurodegenerative illnesses later in life [2,3,4]. Clinical investigations have revealed how exposure to environmental stressors early in life and throughout the developmental years of childhood and adolescence results in resilient or maladaptive behavior with a significant impact on cognition later in life [5]. Multiple types of stressors (multi-hit), such as exposure to prenatal and postnatal malnutrition [6,7], viral and bacterial infections, trauma, and social maltreatment and neurotoxins [8,9] act in variable combinations and profoundly disrupt brain development, thus may considerably increase the pathophysiology of neuropsychiatric disorders in affected individuals. Moreover, maternal inflammatory reactions during pregnancy [10], antenatal infection in preterm newborns [11] and neonatal infection [12] have also been linked to a lower intelligence quotient in offspring and increased chances of developing neuropsychiatric disorders such as schizophrenia and depression [13]. A combination of genetic, physical and environmental factors are known to interact to cause schizophrenia, resulting in the development, maintenance and evolution of the disease symptoms, but determining the exact origin of the disease is challenging [14,15].

Epidemiological data suggest that maternal malnutrition, a type of nutritional stress, has long been associated with changes in neurodevelopment, physical growth indices and brain structure [16,17]. Experimental malnutrition has been variously reported to affect the genesis, migration, plasticity and differentiation of neurons and glial cells during the critical periods of brain development [18,19,20,21,22,23], which are key factors in impaired neurocognitive development. Protein malnutrition has also been linked to an increased risk of neurodevelopmental and metabolic abnormalities in offspring, including autism spectrum disorder (ASD), attention deficit hyperactivity disorder (ADHD) and schizophrenia [24,25].

Experimental studies have also revealed behavioral and cognitive deficits, as well as physical development retardation and poor motor coordination in animal models following exposure to stressors such as maternal and early life infections, neurotoxicants, perinatal protein and protein-calorie malnutrition [22,26]. Perinatal protein malnutrition is also known to cause low anxiety in rodents, and such effects are directly proportional to the length of the period of malnutrition, and the longer exposure to a deficient diet results in long-lasting anxiolytic-like effects [18,27].

Moreover, the unhealthy lifestyles in which the children grow lead them to encounter viral and bacterial infections, which are common problems in already malnourished children, especially those from a low socio-economic background due to the compromised immune system [28]. Perinatal immune activation by viral and bacterial infections negatively impacts ongoing brain development and may enhance the risk of developing neuronal and other developmental disorders [4,29,30]. Lipopolysaccharide (LPS) is a Gram-negative bacterial cell wall component known to mimic bacterial infection in experimental conditions [31,32]. LPS-induced peripheral infection signals to the brain and causes microglial activation and extensive production of inflammatory cytokines, leading to behavioral pathology and cognitive impairment [33,34]. Prenatal exposure to LPS has been reported to alter developmental trajectories in neuron–microglia communication in the brains of young offspring, along with the imbalance in the offspring’s immune system. These prenatally exposed animals exhibit behavioral changes resembling schizophrenia-like phenotype in adulthood [35]. LPS-induced inflammation has also been associated with increased chances of anxiety disorders [36] and major depressive disorders [37].

Deltamethrin (DLT), a type II synthetic pyrethroid insecticide (cyano group at carboxyl α-position), has been largely used in agricultural, medical and domestic applications all over the world for more than 30 years [38,39]. Epidemiological data suggest a clear link between early life exposure to pyrethroid insecticides and neurodevelopmental disorders, viz., ADHD, ASD and developmental delay [39,40]. Pyrethroid exposure poses a public health concern, and the neurobehavioral performance deficiencies include decreased motor activity [41,42,43]. Occupational pesticide exposure has also been related to Parkinson’s disease (PD) [44], Alzheimer’s disease (AD) [45], amyotrophic lateral sclerosis (ALS) [46], dementia and cognitive impairments in several other diseases [47]. Other functions affected include delayed perceptual thinking and linguistic comprehension levels in children, as well as early growth and development impairments [48].

In this study, protein malnutrition (PMN) has been used as the prime stressor, and the F1 pups born to such nutritionally challenged dams were exposed to additive stressors, i.e., LPS and DLT, keeping in view their high susceptibility to infections and other environmental exposures. Thus, the concept of the multi-hit in the current context arises because of the susceptibility of early life development to numerous stressors, acting synergistically during perinatal life.

Thus, it is crucial to understand how the additive, synergistic or cumulative interplay of protein malnutrition, bacterial infection and a neurotoxicant pesticide affect behavioral development during adolescence and adulthood in terms of locomotor activity and anxiety status. Additionally, it was essential to establish whether there was any relationship between exposure to such stressors and the development of neurological disorders.

## 2. Materials and Methodology

Wistar albino rats used in the study were maintained under a standard animal house facility of the School of Studies in Neuroscience, Jiwaji University, Gwalior, in groups of 3 rats/cage (polypropylene cages; 52 cm × 28 cm × 22 cm) with clean and dust free husk bedding. The animals were maintained at a controlled temperature (25 ± 2 °C) and humidity (50–65%) with a fixed 12:12 h light–dark cycle and ad libitum food and water. The maintenance of the animal house was carried out as per the approved conditions and requirements of the Institutional Animal Ethics Committee of Jiwaji University, Gwalior (M.P), India.

Three-month-old virgin female rats (n = 32; body weight 140–150 gm) were selected from the breeding colony and shifted to control (20% protein; n = 16) and LP (8% protein; n = 16) diets for 15 days prior to mating to achieve a state of PMN before conception. Following acclimatization to the diet regimes, the females were placed for mating with healthy male rats in a ratio of 2:1. The pregnancy was confirmed through a vaginal smear test followed by increased body weights. Males were separated, and the pregnant females were maintained on their respective diets throughout gestation and lactation. The protein-malnourished mothers were equally fertile with no issues of pregnancies and mortality, except that the litter size was low. The day of birth was marked as postnatal day 0 for F1 pups. Both litter weight and size were recorded, and the pups were properly monitored for any abnormality and reared with their mothers in aseptic conditions. Litter size was adjusted to eight per dam to prevent dissimilarity among the groups due to different litter sizes. Both the control and LP diets were designed and procured from the National Institute of Nutrition (NIN) Hyderabad, India. The diet composition is tabulated as a Appendix A.

### 2.1. Experimental Groups

The F1 pups obtained from both control and LP mothers were further grouped into eight groups (equal number of males and females), depending on the type of exposure (Figure 1).

Control, Control+LPS, Control+DLT and Control+DLT+LPS groups:

Control group: F1 pups (n = 12; from four different dams) born to females fed with a 20% protein diet were considered as controls and used to assess various behavioral abilities at the ages of 1, 3 and 6 months.

Control rats exposed to bacterial mimetic LPS (Control+LPS group): A set of control F1 pups (n = 12; from four different dams) were exposed to LPS at the dose of 0.3 mg/kg body weight intraperitoneally on PND 3 followed by a booster dose on PND 5 and used to assess various behavioral abilities at the ages of 1, 3 and 6 months.

Control rat pups exposed to deltamethrin (Control+DLT group): Another set of control F1 pups (n = 12; from four different dams) were exposed to DLT at the daily dose of 0.7 mg/kg body weight, intraperitoneally from PND 1 to 7 and used to assess various behavioral abilities at the ages of 1, 3 and 6 months.

Control F1 rat pups exposed to both LPS and DLT (Control+DLT+LPS group): Another set of control F1 pups (n = 12; from four different dams) were exposed to both DLT and LPS at the above specified time points and doses and used to assess various behavioral abilities at the ages of 1, 3 and 6 months.

Protein-malnourished (PMN) groups:

PMN (LP) group: LP F1 pups (n = 12; from four different dams) born to females fed with an 8% protein diet and maintained on the same diet throughout the experimental period were considered as LP/PMN group. These LP F1 rats were maintained on an LP diet and assessed for various behavioral abilities at the ages of 1, 3 and 6 months.

LP F1 rat pups exposed to bacterial mimetic LPS (LP+LPS double-hit group): LP F1 pups (n = 12 from four different dams) were exposed to LPS at the dose of 0.3 mg/kg body weight, intraperitoneally on PND 3 followed by a booster dose on PND 5. These LP+LPS rats were maintained on an LP diet and assessed for various behavioral abilities at the ages of 1, 3 and 6 months.

LP F1 rat pups exposed to deltamethrin (LP+DLT double-hit group): Another set of LP F1 pups (n = 12; from four different dams) were exposed to DLT at a daily dose of 0.7 mg/kg body weight intraperitoneally from PND 1 to 7. These LP+DLT rats were maintained on an LP diet and assessed for various behavioral abilities at the ages of 1, 3 and 6 months.

LP F1 rat pups exposed to both LPS and DLT (LP+DLT+LPS multi-hit group): A third set of LP F1 pups (n = 12; from four different dams) were exposed to both DLT and LPS at the above specified time points and doses. These LP+DLT+LPS rats were maintained on an LP diet and assessed for various behavioral abilities at the ages of 1, 3 and 6 months.

Both LPS (E. coli, serotype O11:B4) and DLT (D9315-10MG) were procured from Sigma Aldrich. The LPS solution was prepared in sterilized phosphate-buffered saline (PBS), while DLT was dissolved in dimethyl sulfoxide (DMSO).

To perform low volume and error-free injection of DLT and LPS at a steady flow rate of 30 µL/min and to confirm absolute absorption, Stoelting Nanoinjector and Hamilton micro-syringe were used under hygienic settings. To counter any bias, control animals were injected with the vehicle alone. No anesthetic procedure was followed during the injection; pups were instead gently handled during the delivery of the injection and then quickly transferred to their respective dams to minimize the separation stress from the mother. To overcome sex-specific differences, an equal number of males and females were used in the study as post hoc analysis did not bring any sex-specific variations during data analysis.

### 2.2. Behavioral Studies

Specific behavioral abilities were assessed in F1 generation rats (n = 12/group/time point, consisting of 6 males and 6 females from different dams) of all the groups (Control, LP, Control+LPS, LP+LPS, Control+DLT, LP+DLT, Control+LPS+DLT and LP+DLT+LPS) at the ages of 1, 3 and 6 months.

#### 2.2.1. Open Field Test

To quantify locomotor activity in F1 rats and track the animal path, the Autotrack system (Columbus Instruments, North Hague Avenue, Columbus, OH, USA) was used. The system tracks the motion of the animal with the help of infrared photocells placed around the open field arena made from a transparent acrylic box (43 × 43 × 22 cm) connected to the computer interface with preinstalled Optovarimax Autotrack software version 4.4. The test animals were subjected to the testing cage of the activity monitor 5 min prior to the final data recording to avoid novelty-induced exploration, followed by a recording of the locomotor behavior for a 20 min test session. The test arena was cleaned with 70% alcohol after every trial to avoid interference from animal odors, i.e., urination and defecation due to the presence of pheromones. The parameters recorded include the total distance traveled (DT), resting time (RT), stereotypic time (ST), ambulatory time (AT), horizontal count (HC) and center zone activity (center zone entries and time) as indicative of spontaneous motor activity as described previously [22,30,49]. From each open field track report, time in the square analysis was conducted, and animals spending more time in the center zone was considered indicative of low anxiety. The amount of time an animal spends engaging in stereotypic behavior, such as abnormal repeated scratching, rearing and grooming that breaks a single optical beam, is known as stereotypic time. To minimize the circadian cycle effect on open-field behavior, testing of all the animals was performed between 9:00 a.m. and 5:30 p.m.

#### 2.2.2. Elevated Plus Maze

To assess the anxiety and fear behavior in F1 generation rats, an elevated plus maze elevated 50 cm above the base, consisting of two opposite closed arms (with 15 cm high walls) and two opposite open arms (without walls), was used. This formed a setup of four arms originating from a common point called the center zone. Before starting the test, the rats were acclimatized for 45 min in the experimental room. For testing, the animal was placed in the center zone facing one of the open arms, and exploratory behavior was recorded by a preinstalled real-time video tracking system camera placed above the apparatus and attached to a computer with ANY-maze software (Stoelting, Wood Dale, IL, USA, v4.82) for 2 min with three trials/animal with a gap of 2 h between trails. The parameters of time spent and number of entries to the open and closed arms [22,30,49], center zone time and entries, and total open arm entries were automatically recorded and tracked by the ANY-maze software. The data were finally analyzed and expressed as percent time spent, percent entries in the open arms, time and entries in the center zone, and total open arm entries. All the procedures were performed between 9:00 a.m. and 5:30 p.m. to minimize the influence of possible circadian changes on the behavior. The percent time spent and the number of entries (%) in open arms were calculated by the following formula:time spent in open arms/total test time × 100; entries in open arms/total entries × 100, respectively. 

#### 2.2.3. Light and Dark Test

To assess the anxiety- and depression-like behavior in F1 rats, a light and dark test was performed. This test is used to analyze the animal’s reaction to the aversion and anxiety-prone context of being in brightly illuminated areas. The apparatus consists of two compartments; the light compartment is 2/3 of the box and is brightly lit and open, while the dark compartment is 1/3 of the box and is covered and dark. Both the compartments are connected by a small doorway with an opening of 7 cm diameter. Rats were placed in the dark compartment and allowed 5 min to explore freely. To avoid a bias produced by novelty-induced exploration, the first 1 min of data was excluded from the analysis. To record and analyze the time spent and the number of entries in each chamber [49], an overhead real-time video tracking system camera was mounted on the ceiling focused on the middle of the apparatus box and attached to the computer set up with preinstalled ANY-maze software version v4.82 (Stoelting, Wood Dale, IL, USA). The apparatus’s inner arena was cleaned with 70% alcohol after each animal testing to avoid biasing with the previously tested animal left due to urination and defecation. The data were finally analyzed and expressed as percent time spent and entries in the light chamber. All the procedures were performed between 9:00 a.m. and 5:30 p.m. to minimize the influence of possible circadian changes on the behavior. The percent time spent and number of entries (%) in the light zone were calculated by the following formula:time spent in the light zone/total test time × 100; entries in light zone/total entries × 100, respectively.

#### 2.2.4. Rotarod Test

A rotarod was used to assess the neuromuscular coordination in terms of motor coordination and motor learning in F1 generation rats as their ability to balance on an accelerating rotating rod. The apparatus consists of a 40 cm long rubber-coated horizontal metal rod (3-inch diameter) attached to a motor through which the speed of the rotating rod is maintained. It is divided into 4 sections, separated by 10 cm long plastic discs to enable testing of 4 animals at once. The rotarod was operated with a computer interface installed with Rotamex-5 version 1.2.3 software. The activity of the rat was detected with the infrared sensors (emitter and detector) attached to the instrument as the rat ran on the rotating rod with any particular rpm selected from the software. The test procedure consisted of two periods. The first period was acclimatization, in which rats were acclimatized for 3 days with three trials per day per animal (start speed: 2 rpm; maximum speed: 8 rpm; duration: 100 s). Another period was the retention period/final day test, which was performed after 24 h of last acclimatization with a starting speed of 2 rpm and a maximum speed of 40 rpm for 420 s. The latency to fall (off) from the accelerating rod (during the provided experimental time) was automatically detected by the infrared sensors and recorded using the Rotamex software [49]. All the animals were tested on the same day from 9:00 a.m. to 5:30 p.m. in order to minimize the influence of possible circadian changes on the behavior.

#### 2.2.5. Statistical Analysis

Results were expressed as the standard error of the mean (±SEM). The standard statistical software Sigma Stat 13.5 was used for all the statistical analyses. The data were analyzed using one-way (ANOVA) for (group-wise comparison) and two-way (ANOVA) (for comparison between groups with two independent variables, i.e., diet and infection) followed by post hoc Holm–Sidak test. The significance level was set at a *p*-value of ≤0.001 for highly significant, and ≤0.05 was considered significant.

## 3. Results

### 3.1. Physical Development

No statistically significant difference was noticed in the appearance of developmental or physical landmarks, including ear pinna detachment, eye opening, incisor eruption, vaginal opening and testes descent amongst all the eight groups. However, LP and LP exposed group animals (LP+LPS; LP+DLT; LP+DLT+LPS) displayed drastic fur loss with stunted body growth, whereas the control group and control-treated animals revealed healthy physical appearance in terms of both hair and body growth.

#### Body Weight (gm)

The F1 offspring of LP mothers exhibited a clear compromise in physical growth, with lower body weight with age as compared to the controls. Furthermore, exposure with either LPS or DLT (LP+LPS and LP+DLT) or cumulative exposure of both LPS and DLT to LP rats (LP+DLT+LPS) showed further significant reduced body weight at the age of 1 and 3 months as compared to age-matched control and LP alone animals. However, LP+DLT+LPS rats also revealed significantly reduced body weight as compared to age-matched control and control-treated rats at 6 months of age. Furthermore, the body weight of the LP-treated group rats also remained significantly low as compared to their corresponding age-matched control-treated groups (Figure 2A–E). The statistical mean and f-values, along with the level of significance, are given in Table 1 and Table 2.

### 3.2. Double-Hit and Multi-Hit Exposure of Lipopolysaccharide and Deltamethrin Induced Hyperactivity and Low Anxious Behavior in Protein-Malnourished Rats

Open field analysis demonstrated that maternal protein malnourishment resulted in hyperactivity and low anxiety-like behavior, as revealed by significantly increased DT (Figure 3B), ST (Figure 3C), AT (Figure 3D), HC (Figure 3E), center zone entries (Figure 3F), center zone time (Figure 3G) and a significantly decreased resting time (Figure 3A) in LP rats at the age of 1, 3 and 6 months as compared to the age-matched control groups showing the effects of maternal protein malnutrition. Exposure with either LPS or DLT (LP+LPS and LP+DLT) or cumulative exposure of both LPS and DLT to LP rats (LP+LPS+DLT) resulted in further exaggeration in the hyperactivity and low-anxiety behavior with a highly significant increase in DT, AT, ST and HC at the age of 1, 3 and 6 months as compared to the age-matched control and LP alone animals. Such hyperactivity and low anxiety behavior were also observed in control animals exposed to either LPS or DLT or both LPS+DLT, but the mean values for DT, AT, ST and HC remained comparatively low as compared to their respective LP groups. Time in square analysis from open field tracks showed that LP multi-hit rats (LP+DLT+LPS) spent significantly more time in the center zone with increased center zone entries as compared to age-matched control and LP-alone animals. The locomotive hyperactivity behavior was highly pronounced at 3 months of age, and the animals remained hyperactive and anxious throughout their life, i.e., by 6 months of age studied in this investigation.

Open field track reports also confirmed that the above data showed a low anxiety-like behavioral profile and locomotive hyperactivity in LP and LP-treated animals compared to age-matched controls, which showed normal exploratory behavior (Figure 4a,e,i). Combo exposure of LPS and DLT to both control and LP group animals further reduced the anxiety levels (Figure 4b–d,f–h,j–l) comparatively more in LP-treated animals (Figure 4n–p,r–t,v–x), with very frequent haphazard center zone arena exploration. Furthermore, multi-hit animals showed drastic behavioral impairments accompanied by irregular track activity with maximum periphery exploration/wall clinging called thigmotaxis and increased time spent in the center zone and corners, showing severe hyperactivity or low anxiety and fearless behavior. Additionally, the dense area in the center zone depicts stereotyped repetitive rearing and horizontal back-and-forth movements revealed by multi-hit animals, further suggesting their low-anxiety profile. The statistical mean and f-values, along with the level of significance, are given in Table 3 and Table 4.

### 3.3. Elevated Plus Maze Test (EPM) Revealed Low Anxiety Phenotype in Protein-Malnourished Rats Treated with Lipopolysaccharide and Deltamethrin

The elevated plus maze (EPM) test also confirmed the low anxiety behavior following LPS and DLT exposure to LP rats. Data analysis from the EPM test revealed that LP animals spent significantly more time in open arms (% open arm time; Figure 5B) along with a significantly increased number of open arm entries (% open arm entries) (Figure 5A). In addition, both the time spent in the center zone and center zone entries (Figure 5C,D) were also significantly more in LP animals. However, upon single or combined treatment of LPS and DLT to LP animals, there was a further significant increase in both % open arm entries and % time spent in open arms, as well as the time spent in center zone and center zone entries with the maximum increase in the multi-hit treatment group (LP+LPS+DLT), demonstrating highly hyperactive and low anxiety-like phenotype as compared to age-matched control animals. The control animals also responded similarly to LPS or DLT or LPS+DLT treatments with a significant increase in % open arm entries and % open arm time as well as time in center zone and center zone entries at 1, 3 and 6 months, but to a lower degree as compared to their corresponding LP group rats. The number of open-arm entries increased gradually, with the complexity of exposure being highest in the LP multi-hit group (Figure 5E). Such behavior persisted consistently until the age of 6 months, as studied in this investigation. The data showing the mean and the f-values are tabulated in Table 5 and Table 6.

Elevated plus maze track reports also confirmed the above data showing the low anxiety-like behavioral profile in LP and LP-treated animals as compared to the respective age-matched control, which stays away from the open arms edges (Figure 6a,i,q). Combo exposure of LPS and DLT to both control (Control+LPS; Control+DLT; Control+DLT+LPS) and LP (LP+LPS; LP+DLT; LP+DLT+LPS) group animals further reduced the anxiety levels (Figure 6b–d,j–l,r–t), showing highly anxious profile in LP treated animals (Figure 6f–h,n–p,v–x), with very frequent open arm and center zone entries. Moreover, the multi-hit group rats frequently explored the open arms until the extreme distal ends, depicting hyperactivity or low-fear behavior (Figure 6d,l,t,h,p,x).

### 3.4. Light and Dark Box Test also Revealed Low Anxiety Behavior in Protein-Malnourished Rats Treated with LPS and DLT

OFT and EPM results were further supported by the light and dark box test. Data assessment from histograms of LD box revealed that LP animals showed significantly increased % light zone entries (Figure 7A) and % light zone time at 1, 3 and 6 months (Figure 7B), which indicates the low anxiety-like behavior with an increased tendency to explore the light zone in contrast to the biased normal nocturnal behavior of rats shown by age-matched control animals, revealing the impact of maternal protein malnutrition. Further exposure of LPS and DLT or both LPS+DLT to the control and LP F1 rats caused a sharp increase in both the % light zone entries and % light zone time, suggesting the impact of such double or multi-hit exposure in inducing hyperactivity and low anxiety. However, these changes were significantly higher in the LP multi-hit treated group, which indicates their higher susceptibility to developing such phenotypes.

Light and dark test track reports of animals also supported the above data, showing low anxiety profiles in LP and LP-treated animals as compared to age-matched control animals (Figure 7C(a,i,q)). Combo or multi-hit exposure of LPS and DLT in both control (Control+LPS; Control+DLT; Control+DLT+LPS) and LP (LP+LPS; LP+DLT; LP+DLT+LPS) group animals further showed reduced anxiety levels (Figure 7C(b–d,j–l,r–t)), comparatively more in LP treated animals (Figure 7C(f,g,n,o,v,w)) with irregular, random locomotor activity due to frequent entries into light compartment. Additionally, double or multi-hit animals depicted immobility at the doorway opening towards the light compartment, thus suggesting freezing behavior (Figure 7C(c,f,h,l,p,s,t,w,x). The statistical mean and f-values, along with the level of significance, are tabulated in Table 7 and Table 8.

### 3.5. Cumulative Exposure of LPS and DLT to Protein-Malnourished (Multi-Hit) Rats Resulted in Hyperlocomotion and Motor Stereotypy Phenotype

The rotarod test was used to evaluate motor coordination. Interestingly, LP animals showed significantly increased latency to fall off from the accelerating rod of the rotarod at the age of 3 and 6 months as compared to age-matched control animals, indicating increased stereotyped behavior in LP animals, which tend them to keep moving purposelessly. Upon single or combined exposure of LPS and DLT to LP animals (LP+LPS, LP+DLT and LP+DLT+LPS), there was a further significant increase in latency to fall both at the age of 3 and 6 months. However, the multi-hit (LP+DLT+LPS) group rats were able to retain themselves for the maximum time on the accelerating rod, showing the highest latency to fall off both at 3 and 6 months of age, which reveals increased hyperactivity and motor stereotyped behavior when compared to the age-matched control group. Such behavioral deficits were also significantly observed in Control+LPS, Control+DLT and Control+DLT+LPS groups at 3 and 6 months of age when compared to age-matched normal control group. Such significantly higher retention time on rotating rotarod by the double and multi-hit group rats indicate increased hyperactivity and motor stereotyped behavior. The statistical mean and f-values, along with the level of significance, are mentioned in Figure 8 and Table 9 and Table 10.

## 4. Discussion

Perinatal stress has been widely documented as a contributing factor to several neurodevelopmental disorders, such as schizophrenia, ADHD, AD, PD and many others [50,51,52]. Many research groups have focused on the two-hit or dual-hit hypothesis, which postulates that an early genetic predisposition during critical developmental periods with a second hit, typically an environmental insult, such as an infection, nutritional deficiency or exposure to neurotoxicants, increases the likelihood of development of schizophrenia [30,53,54].

Even in the absence of genetic predisposition, perinatal life is inherently vulnerable to various environmental insults. Consequently, the presence of multiple stressors during early life, beyond the scope of the dual hit hypothesis, significantly increases the risk of schizophrenia and other neuropsychiatric disorders. In this study, we investigated the impact of multiple stressors during development and early life on the development of behavioral traits similar to schizophrenia and other psychotic disorders.

The severity of neurological problems is directly associated with the loss of body weight of an individual as it can increase the risk of systemic infections and may also raise the chances of morbidity and mortality in later life [55,56]. In this study, we recorded a persistent and significant loss of body weight in LP and LP-treated double-hit and multi-hit group rats as compared to the respective control rats. Such a significant loss of body weight may be due to the persistent overactivation of hypothalamic–pituitary–adrenal (HPA) axis as a response towards early life stressors [57], contributing to the development of metabolic dysregulation [58], as reported both in animal models [59] and human subjects [60]. In addition, the earlier reports from our lab also support that protein deprivation causes emaciation due to shrinkage of body size [22,23]. The available literature suggests that there is a direct link between the immune system and brain development. Both the brain and the immune system are not fully formed at birth but rather continue to mature in response to the postnatal environment, thus making them susceptible to early life stressors [61]. Studies in both humans [62] and animal models [63,64,65] suggest that a drop in body weight during early neonatal life following infection is an indication of sickness behavior. The population-based birth cohort studies have also found a link between low birth weight and many neuropsychological deficits, such as schizophrenia [66,67] and ADHD [68,69]. All these studies suggest that compromised physical growth and immune system following multi-hit exposure may predispose individuals to develop neuropsychiatric deficits.

To further investigate how the cumulative interplay impact of all these stressors affects the behavior of multi-hit rats, a battery of behavioral tests was used. The open field test, frequently used to assess rodent locomotion, anxiety and stereotyped behavior, including grooming and rearing [70,71], demonstrated that maternal PMN results in hyperactivity and stereotyped behavior and a low anxiety-like behavioral phenotype, as assessed by a significant increase in DT, ST, AT, HC, center zone entries and center zone time. In addition, there was a significant decrease in RT, suggesting restless behavior. Such changes were further pronounced following exposure to LPS or DLT or additive exposure of both LPS+DLT to LP F1 rats, showing severe behavioral deficits with frequent central zone exploration as shown by an increase in center zone entries and time, depicting hyper-locomotion and spontaneous jumping against the wall of the open field arena or wall-hugging as a stereotyped behavior, which persisted through adolescence and adulthood. Increased center zone exploration is an indicator of low anxiety as the rodents have a natural tendency to avoid the open center [72], while the increased horizontal and vertical counts suggest increased exploratory activity [73,74]. OFT data also revealed that LP F1 rats exposed to both LPS and DLT spent more time in stereotypic activities like scratching; licking of the head, neck and trunk; and other grooming behaviors. Such behaviors were persistent throughout adolescence and adulthood. Stereotypies or repetitive behavior are the repetitions of certain motor patterns with no apparent goals or functions, which coincides with the theory of being unable to translate cognition into actions. Thus, LP multi-hit rats lose self-control mechanisms such as control of attention or emotions because they have lost the chain between knowledge and actions and lack the capacity to refrain from acting spontaneously [75,76]. In addition, the hyperactivity and thigmotaxis also indicate the stereotypic activity traversing the same locomotor trajectory repeatedly [77]. Although the equivalence of rodents and human repetitive behavior is variously argued, the probability of repetitive behavior and increased self-grooming in rodents is directly correlated with hyperactivity and stereotypy and may indicate spontaneous aggression, which has long been linked with obsessive–compulsive disorder (OCD). Grooming in rodents has become a useful strategy for modeling different mood and psychiatric disorders and understanding neural circuitries underlying complex motor patterns [78]. Thus, the increased self-grooming in rodents indicating complex repetitive, self-directed and sequentially patterned behaviors may correlate indirectly to human brain disorders, including chain of motor actions and complex patterning of motor activities [78]. The anxiety-like states also alter rodent self-grooming and its sequencing [79,80]. The increased self-grooming in rodents has been modeled with disease symptoms in OCD [81,82], ASD [83,84], anxiety and panic disorders, and schizophrenia [85,86]. Interestingly, the LP F1 rats, from early adolescence through late adulthood, exhibited increased central time duration in addition to overall increased locomotion, indicating hyperactivity, motor stereotypy and low anxiety symptoms that are typical of both ADHD [87] and schizophrenic [88,89] patients. This suggests that increased movement and stereotypic behavior in LP and LP multi-hit group rats is indicative of their inability to adapt to novel conditions or signs of inadequate habituation [90,91]. When tested with rotarod, these single and multi-hit animals maintained themselves on the accelerating rod of the rotarod for a longer time, showing significantly increased latency to fall off when compared with the control animals. This indicates their motor stereotypy behavior that makes them keep moving abnormally and purposelessly. The motor stereotypies have been described as complex, repetitive, rhythmic, often bilateral movements with a typical onset in early childhood [92]. Such movements in the multi-hit rats may be the resultant of hyperactivity, low anxiety and fearless behaviors as revealed by open field test, elevated plus maze, and light and dark test. Over time, the emotional responses may become dissipated and replaced by hyperactivity and stereotyped responses. In this study, motor stereotypy refers to an excessive repetition of one type of motor response [93], i.e., prolonged maintenance on the rotating rod of the rotarod and increased latency to fall off by multi-hit animals. Such exaggerated hyperactivity and repetitive and stereotyped behavior also indirectly indicate an inattention behavior, as rats are reluctant to engage in tasks requiring sustained attention. Such behavior shown by multi-hit rats is also seen in many neuropsychiatric conditions, such as schizophrenia [94,95], OCD [96], autism spectrum disorder (ASD) and ADHD [97,98]. Recently, exposure to DLT alone has been linked to long-term neurobehavioral deficits in rodents when exposed postnatally [40] or in offspring when exposed during pregnancy [99]. Locomotor hyperactivity, attention deficits and elevated dopamine transporter and receptor levels commonly reported following DLT exposure are also typically seen in children with ADHD [99]. Studies examined with exposure to maternal protein deficiency or newborn viral or bacterial infections have similarly demonstrated impairment in behavior as measured by the OFT paradigm [22,23,30,100,101] consistent with what was observed in the present study.

These OFT results were further supported by EPM data, another commonly used test to assess anxiety-like phenotype, as rodents show an innate fear of height, which conflicts with curiosity and a drive for spontaneous exploration [102,103]. A higher degree of anxiety is often indicated by more frequent entries and time spent in the closed arms of EPM, while the higher open-arms exploration suggests anxiolytic behavior [71,102,104]. Rodents naturally favor closed arms and, in a typical 10 min experiment, spend the majority of their time in the closed arms [104,105]. The multi-hit animals tested in the EPM spent the most time in open arms with significantly increased % time and % entries along with significantly more center zone exploration, indicating hyperactive, low anxiety and fearless phenotype. Such changes increase gradually with the complexity of exposure and age, revealing that perinatal stressors lead to prolonged and consistent behavioral changes through adolescence to adulthood, as also seen in most schizophrenic patients. Low anxiety phenotype was also reported following exposure to viral and bacterial infection in rats [30] and humans [106,107], mouse models of tauopathy [108] and exposure to bisphenol A [109]. Unconditional anxiety responses were tested in rats by light and dark zone exploration tests [110]. Similar to the EPM, the rats were placed in an open arena that included protected (dark compartment) and unprotected (bright compartment) areas with free access between the two [111]. Most rodents exhibit a natural affinity for protected, dark spaces [71,106,112]; thus, increased avoidance of the light chamber is interpreted as increased anxiety [113]. However, increased time spent in the light zone or shorter entry delays are interpreted as signs of anxiolytic behavior or low anxiety-like phenotype [110,114]. Light and dark box tests also revealed hyperactivity and low anxiety-like behavior as multi-hit LP rats showed a significant increase in light zone exploration with increased % light zone entries and % light zone time. The increased time spent in the light zone or shorter entry delays are interpreted as signs of anxiolytic behavior or low anxiety-like phenotype [110,114].

The present study thus suggests that perinatal additive/cumulative multi-hit exposure of PMN, LPS and deltamethrin during early life leads to long-term deficits in terms of decreased bodyweight, hyperactivity, motor stereotypy and low anxiety, typically seen in patients with neuropsychiatric disorders like schizophrenia and ADHD. Although the heritability of ADHD and schizophrenia is quite high but not absolute [115,116], environmental influences during prenatal and postnatal development might also be playing a substantial role in the pathogenesis of these diseases. The results obtained in this study thus suggest that the cumulative impact of early life stressors may act as crucial risk factors and may pre-dispose individuals to develop such psychopathologies during adolescence and adulthood.

## 5. Conclusions

The present results indicate that additive exposure to multiple perinatal stressors during early life may lead to severe behavioral abnormalities in terms of hyperactivity, motor stereotypy and low-anxiety phenotype in adulthood, which may mimic the pathophysiology of neuropsychiatric disorders like ADHD and schizophrenia and may enhance the risk of developing such psychopathologies in adulthood.

## Figures and Tables

**Figure 1 brainsci-13-01360-f001:**
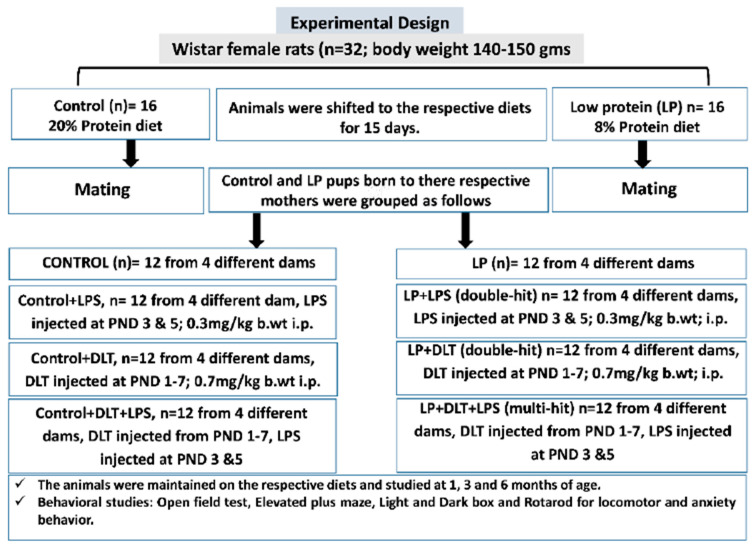
Overview of the experimental design and procedures followed.

**Figure 2 brainsci-13-01360-f002:**
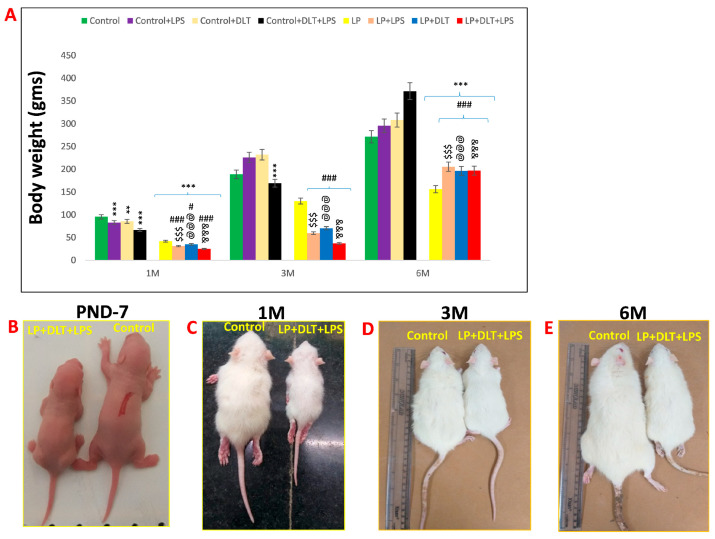
Effects of singular or cumulative exposure of LPS and DLT to maternally PMN F1 rats on physical development: LP group animals showed significantly reduced body weight as compared to age-matched control group. However, LP animals, when treated either with LPS (LP+LPS) or DLT (LP+DLT) or both (LP+DLT+LPS), revealed further reduction in body weight as compared to their respective control groups (n = 12/group; (**A**)). (**B**–**E**) reveals the difference in the body appearance of multi-hit rats and their age-matched controls at the age of PND7, 1 M, 3 M and 6 M. Data were analyzed by one-way (comparison with control) and two-way (comparison between diet and infection) ANOVA and presented as mean ± SEM (n = 12); ** *p* ≤ 0.01, *** *p* ≤ 0.001, for comparison between control group and treated animals; ^#^
*p* ≤ 0.05, ^###^
*p* ≤ 0.001, for comparison between LP group and treated group; ^$$$^
*p* ≤ 0.001; for comparison between Control+LPS and LP+LPS group; ^@@@^
*p* ≤ 0.001, for comparison between Control+DLT and LP+DLT group; ^&&&^
*p* ≤ 0.001, for comparison between Control+DLT+LPS and LP+DLT+LPS group.

**Figure 3 brainsci-13-01360-f003:**
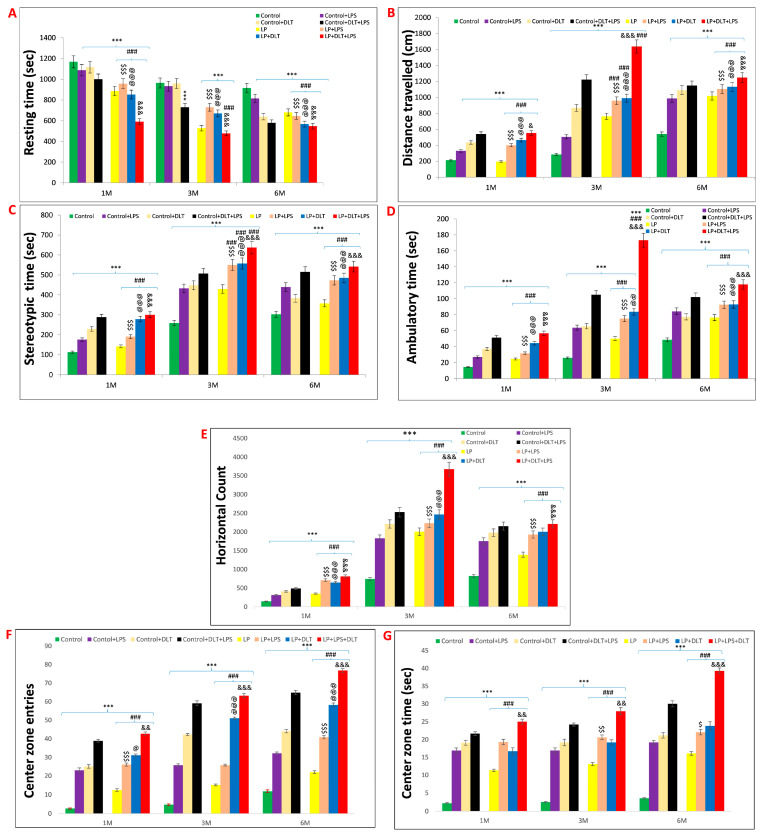
Open field test data showing hyperactivity/inattention and low anxiety-like symptoms in LP+DLT+LPS group rats in adolescence and adulthood: Bar graphs represent resting time (**A**), distance traveled (**B**), stereotypic time (**C**), ambulatory time (**D**), horizontal count (**E**), center zone entries (**F**) and center zone time (**G**) at 1, 3 and 6 months of age. All the treated group rats revealed significantly increased distance traveled, stereotypic time, ambulatory time, horizontal count, center zone entries and center zone time with low resting time when compared to age-matched control. However, LP+DLT+LPS group rats displayed highest level changes in all parameters as compared to control and LP alone rats. Representative open field activity track reports. *** *p* ≤ 0.001, for comparison between control group and treated animals; ^###^
*p* ≤ 0.001, for comparison between LP group and treated group; ^$^
*p* ≤ 0.05, ^$$^
*p* ≤ 0.01, ^$$$^
*p* ≤ 0.001, for comparison between Control+LPS and LP+LPS group; ^@^
*p* ≤ 0.05, ^@@^
*p* ≤ 0.01, ^@@@^
*p* ≤ 0.001, for comparison between Control+DLT and LP+DLT group; ^&^
*p* ≤ 0.05, ^&&^
*p* ≤ 0.01, ^&&&^
*p* ≤ 0.001, for comparison between Control+DLT+LPS and LP+DLT+LPS group.

**Figure 4 brainsci-13-01360-f004:**
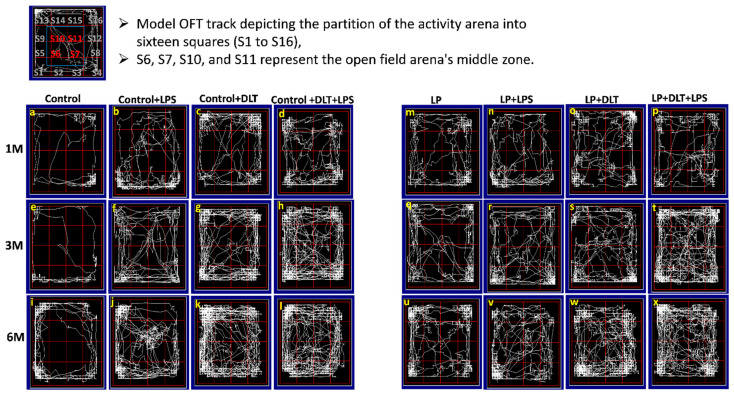
Revealed that control animals safely explore the open arena (**a**,**e**,**i**), while LP F1 rats showed increased exploration (**m**,**q**,**u**) and LP+LPS (**n**,**r**,**v**), LP+DLT (**o**,**s**,**w**) and LP+DLT+LPS (**p**,**t**,**x**) group rats exhibit further behavioral impairments as compared to control+LPS (**b**,**f**,**j)**, control+DLT (**c**,**g**,**k**) and control+DLT+LPS (**d**,**h**,**l**). Moreover, LP+DLT+LPS group rats also present increased and haphazard center zone exploration (S6, S7, S10 and S11). Data were analyzed by one-way (comparison with control) and two-way (comparison between diet and infection) ANOVA and presented as mean ± SEM.

**Figure 5 brainsci-13-01360-f005:**
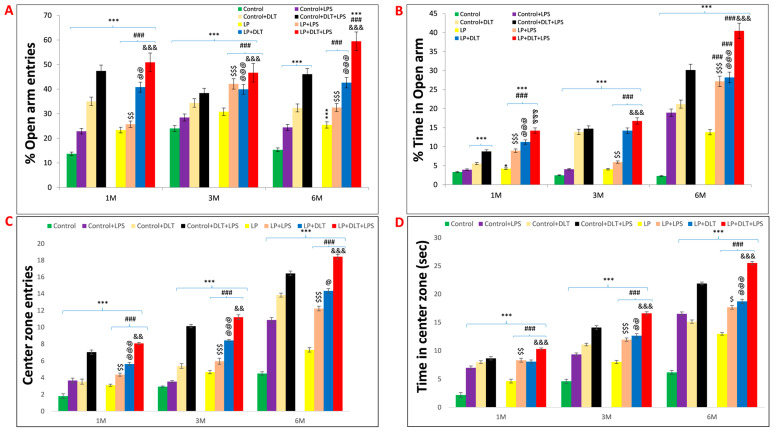
Elevated plus maze data also showed low anxiety, hyperactivity/inattention and low fear-like symptoms in LP+DLT+LPS group rats: Bar graphs representing the percent open arm entries (**A**), percent open arm time (**B**), center zone entries (**C**), center zone time (**D**) and total open arm entries (**E**) at 1, 3 and 6 months of postnatal age. From bar graphs, it was revealed that LP F1 rats prefer open arms and center zone exploration over closed arms, thus showing significantly increasing number of open arm entries and time (**A**,**B**), center zone entries and time (**C**,**D**), and total open arm entries (**E**) as compared to control group. However, LP+DLT+LPS group rats displayed further increase in all parameters as compared to control and LP alone animals. * *p* ≤ 0.05, *** *p* ≤ 0.001, for comparison between control group and treated animals; ^###^
*p* ≤ 0.001, for comparison between LP group and treated group; ^$^
*p* ≤ 0.05, ^$$^
*p* ≤ 0.01, ^$$$^
*p* ≤ 0.001, for comparison between Control+LPS and LP+LPS group; ^@^
*p* ≤ 0.05, ^@@^
*p* ≤ 0.01, ^@@@^
*p* ≤ 0.001, for comparison between Control+DLT and LP+DLT group; ^&&^
*p* ≤ 0.01, ^&&&^
*p* ≤ 0.001, for comparison between Control+DLT+LPS and LP+DLT+LPS group.

**Figure 6 brainsci-13-01360-f006:**
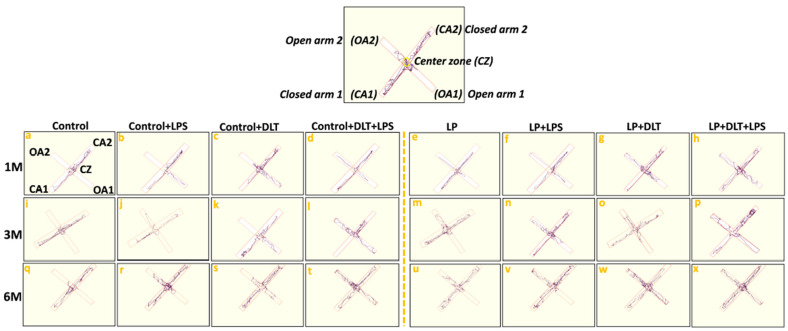
Track records indicate that control group rats behave normally while exploring the EPM arms with little percent open arm entries and time (**a**,**i**,**q**) as compared to LP group animals (**e**,**m**,**u**). However, exposed control+LPS (**b**,**j**,**r**), control+DLT (**c**,**k**,**s**), control+DLT+LPS (**d**,**l**,**t**), LP+LPS (**f**,**n**,**v**), LP+DLT (**g**,**o**,**w**) rats favored exploring open arms over closed ones and center zones. Such behavioral impairment was more severe in the LP+DLT+LPS group animals, showing low anxious profile with very frequent open arm and center zone entries (**h**,**p**,**x**). Moreover, the multi-hit group rats repeatedly explored the open arms to their most distal ends. Data were analyzed by one-way (comparison with control) and two-way (comparison between diet and infection) ANOVA and presented as mean ± SEM.

**Figure 7 brainsci-13-01360-f007:**
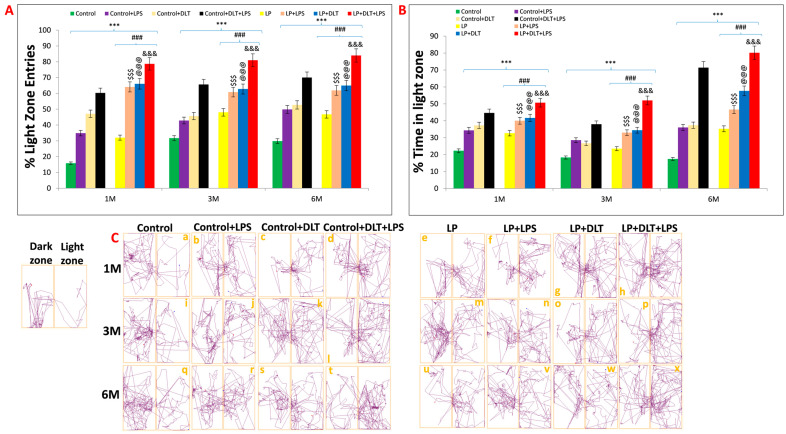
Light and dark box data also indicated low anxiety, hyperactivity/inattention and low fear-like symptoms in LP+DLT+LPS group: Bar graphs representing percent light zone entries (**A**) and percentage light zone time (**B**) at 1, 3 and 6 months of postnatal age: Data analysis showed that LP animals had significantly higher percent light zone entries (**A**) and percent light zone time (**B**) in contrast to age-matched control animals, which preferred dark zone. However, after exposure to either LPS or DLT or both LPS+DLT in control and LP F1 rats, a further significant rise in the percentage of light zone entries and light zone time was shown. Furthermore, such changes were significantly higher in the LP F1 stressed animals with maximum in LP+DLT+LPS group rats. Track records (**C**) also indicate that control group rats showed normal exploratory behavior upon exploring the light and dark zone with few percent light zone entries and time (a,i,q). However, cumulative exposure of LPS and DLT to control and LP F1 group rats revealed further increase in hyperactivity and low anxiety, which was evident by frequent entries into the light compartment. Such behavioral impairment was significantly maximum in LP+DLT+LPS group animals (h,p,x) as compared to age-matched control+LPS (b,j,r), control+DLT (c,k,s), control+DLT+LPS (d,l,t), LP (e,m,u), LP+LPS (f,n,v), LP+DLT (g,o,w) and other groups control Data were analyzed by one-way (comparion with control) and two-way (comparison between diet and infection) ANOVA and presented as mean ± SEM; *** *p* ≤ 0.001, for comparison between control group and treated animals; ^###^
*p* ≤ 0.001, for comparison between LP group and treated group; ^$$$^
*p* ≤ 0.001, for comparison between Control+LPS and LP+LPS group; ^@@@^
*p* ≤ 0.001, for comparison between Control+DLT and LP+DLT group; ^&&&^
*p* ≤ 0.001, for comparison between Control+DLT+LPS and LP+DLT+LPS group.

**Figure 8 brainsci-13-01360-f008:**
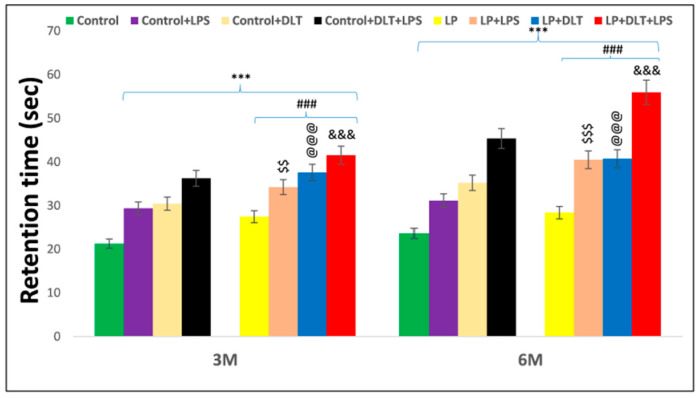
Rotarod performance displayed abnormal stereotypic behavior and inattention/hyperactive-like symptoms in LP+DLT+LPS group. Bar graph indicates the retention time on the accelerating rotating rod at 3 and 6 months of postnatal age. Control and LP F1 group rats exposed to either LPS or DLT or both LPS+DLT showed a significant increase in latency to fall off from the accelerating rotarod, indicating abnormal stereotyped motor behavior, which was more prominent in animals belonging to LP F1 treated groups and LP+LPS+DLT group rats indicated significantly maximum increased latency to fall off among all other groups showing a prominently abnormal stereotypic behavior than age-matched control rats. Data were analyzed by one-way (comparison with control) and two-way (comparison between diet and infection) ANOVA and presented as mean ± SEM; *** *p* ≤ 0.001; for comparison between control group and treated animals; ^###^
*p* ≤ 0.001; for comparison between LP group and treated group; ^$$^
*p* ≤ 0.01, ^$$$^
*p* ≤ 0.001, for comparison between Control+LPS and LP+LPS group; ^@@@^
*p* ≤ 0.001, for comparison between Control+DLT and LP+DLT group; ^&&&^
*p* ≤ 0.001, for comparison between Control+DLT+LPS and LP+DLT+LPS group.

**Table 1 brainsci-13-01360-t001:** Body weight represented as mean values along with level of significance (one-way ANOVA).

	Age	Control vs. LP+LPS	Control vs. LP+DLT	Control vs. LP+LPS+DLT
Body weight (gm)	1 M	[F _(7,95)_ = 25.385, *p* ≤ 0.001]	[F _(7,95)_ = 23.894, *p* ≤ 0.001]	[F _(7,95)_ = 27.811, *p* ≤ 0.001]
	3 M	[F _(7,95)_ = 35.895, *p* ≤ 0.001]	[F _(7,95)_ = 32.917, *p* ≤ 0.001]	[F _(7,95)_ = 42.173, *p* ≤ 0.001]
	6 M	[F _(7,95)_ = 14.956, *p* ≤ 0.001]	[F _(7,95)_ = 17.030, *p* ≤ 0.001]	[F _(7,95)_ = 16.900, *p* ≤ 0.001]

**Table 2 brainsci-13-01360-t002:** Body weight represented as mean values along with level of significance (two-way ANOVA).

	Age	Control vs. LP	Control vs. Control+LPS	Control vs. Control+DLT	Control vs. Control+DLT+LPS	LP vs. LP+LPS	LP vs. LP+DLT	LP vs. LP+DLT+LPS	Control+LPS vs. LP+LPS	Control+DLT vs. LP+DLT	Control+DLT+LPS vs. LP+DLT+LPS
Body weight (gm)	1 M	[F _(1,95)_ = 21.125, *p* ≤ 0.001]	[F _(1,95)_ = 4.952, *p* ≤ 0.001]	[F _(1,95)_ = 3.951, *p* ≤ 0.015]	[F _(1,95)_ = 11.425, *p* ≤ 0.001]	[F _(1,95)_ = 4.261, *p* ≤ 0.001]	[F _(1,95)_ = 2.769, *p* = 0.025]	[F _(1,95)_ = 6.686, *p* ≤ 0.001]	[F _(1,95)_ = 20.433, *p* ≤ 0.001]	[F _(1,95)_ = 19.942, *p* ≤ 0.001]	[F _(1,95)_ = 16.386, *p* ≤ 0.001]
	3 M	[F _(1,95)_ = 16.347, *p* ≤ 0.001]	-	-	[F _(1,95)_ = 5.417, *p* ≤ 0.001]	[F _(1,95)_ = 19.548, *p* ≤ 0.001]	[F _(1,95)_ = 16.569, *p* ≤ 0.001]	[F _(1,95)_ = 25.826, *p* ≤ 0.001]	[F _(1,95)_ = 46.184, *p* ≤ 0.001]	[F _(1,95)_ = 44.881, *p* ≤ 0.001]	[F _(1,95)_ = 36.756, *p* ≤ 0.001]
	6 M	[F _(1,95)_ = 30.977, *p* ≤ 0.001]	-	-	-	[F _(1,95)_ = 11.197, *p* ≤ 0.001]	[F _(1,95)_ = 9.123, *p* ≤ 0.001]	[F _(1,95)_ = 9.252, *p* ≤ 0.001]	[F _(1,95)_ = 24.164, *p* ≤ 0.001]	[F _(1,95)_ = 29.988, *p* ≤ 0.001]	[F _(1,95)_ = 46.891, *p* ≤ 0.001]

**Table 3 brainsci-13-01360-t003:** Locomotor activity assessed by OFT represented as mean values along with level of significance (one-way ANOVA).

	Age	Control vs. LP+LPS	Control vs. LP+DLT	Control vs. LP+LPS+DLT
RT (s)	1 M	[F _(7,95)_ = 28.952, *p* ≤ 0.001]	[F _(7,95)_ = 43.230, *p* ≤ 0.001]	[F _(7,95)_ = 79.067, *p* ≤ 0.001]
	3 M	[F _(7,95)_ = 16.061, *p* ≤ 0.001]	[F _(7,95)_ = 20.097, *p* ≤ 0.001]	[F _(7,95)_ = 33.299, *p* ≤ 0.001]
	6 M	[F _(7,95)_ = 23.689, *p* ≤ 0.001]	[F _(7,95)_ = 30.616, *p* ≤ 0.001]	[F _(7,95)_ = 32.370, *p* ≤ 0.001]
DT (cm)	1 M	[F _(7,95)_ = 30.488, *p* ≤ 0.001]	[F _(7,95)_ = 40.905, *p* ≤ 0.001]	[F _(7,95)_ = 55.327, *p* ≤ 0.001]
	3 M	[F _(7,95)_ = 39.593, *p* ≤ 0.001]	[F _(7,95)_ = 41.488, *p* ≤ 0.001]	[F _(7,95)_ = 79.490, *p* ≤ 0.001]
	6 M	[F _(7,95)_ = 57.726, *p* ≤ 0.001]	[F _(7,95)_ = 60.553, *p* ≤ 0.001]	[F _(7,95)_ = 72.608, *p* ≤ 0.001]
ST (s)	1 M	[F _(7,95)_ = 52.712, *p* ≤ 0.001]	[F _(7,95)_ = 112.616, *p* ≤ 0.001]	[F _(7,95)_ = 127.405 *p* ≤ 0.001]
	3 M	[F _(7,95)_ = 23.211, *p* ≤ 0.001]	[F _(7,95)_ = 23.769, *p* ≤ 0.001]	[F _(7,95)_ = 30.169, *p* ≤ 0.001]
	6 M	[F _(7,95)_ = 35.539, *p* ≤ 0.001]	[F _(7,95)_ = 37.990, *p* ≤ 0.001]	[F _(7,95)_ = 49.767, *p* ≤ 0.001]
AT (s)	1 M	[F _(7,95)_ = 11.878, *p* ≤ 0.001]	[F _(7,95)_ = 20.682, *p* ≤ 0.001]	[F _(7,95)_ = 29.346, *p* ≤ 0.001]
	3 M	[F _(7,95)_ = 9.254, *p* ≤ 0.001]	[F _(7,95)_ = 10.855, *p* ≤ 0.001]	[F _(7,95)_ = 27.597, *p* ≤ 0.001]
	6 M	[F _(7,95)_ = 17.414, *p* ≤ 0.001]	[F _(7,95)_ = 17.573, *p* ≤ 0.001]	[F _(7,95)_ = 27.530, *p* ≤ 0.001]
HC	1 M	[F _(7,95)_ = 63.305, *p* ≤ 0.001]	[F _(7,95)_ = 56.219, *p* ≤ 0.001]	[F _(7,95)_ = 74.260, *p* ≤ 0.001]
	3 M	[F _(7,95)_ = 379.330, *p* ≤ 0.001]	[F _(7,95)_ = 438.713, *p* ≤ 0.001]	[F _(7,95)_ = 746.570, *p* ≤ 0.001]
	6 M	[F _(7,95)_ = 72.442, *p* ≤ 0.001]	[F _(7,95)_ = 78.269, *p* ≤ 0.001]	[F _(7,95)_ = 90.936, *p* ≤ 0.001]
CZE	1 M	[F _(7,95)_ = 19.513, *p* ≤ 0.001]	[F _(7,95)_ = 23.682, *p* ≤ 0.001]	[F _(7,95)_ = 33.439 *p* ≤ 0.001]
	3 M	[F _(7,95)_ = 19.025, *p* ≤ 0.001]	[F _(7,95)_ = 41.747, *p* ≤ 0.001]	[F _(7,95)_ = 52.746, *p* ≤ 0.001]
	6 M	[F _(7,95)_ = 22.483, *p* ≤ 0.001]	[F _(7,95)_ = 35.849, *p* ≤ 0.001]	[F _(7,95)_ = 50.219, *p* ≤ 0.001]
CZT (s)	1 M	[F _(7,95)_ = 18.531, *p* ≤ 0.001]	[F _(7,95)_ = 15.732, *p* ≤ 0.001]	[F _(7,95)_ = 24.720, *p* ≤ 0.001]
	3 M	[F _(7,95)_ = 18.608, *p* ≤ 0.001]	[F _(7,95)_ = 17.114, *p* ≤ 0.001]	[F _(7,95)_ = 26.103, *p* ≤ 0.001]
	6 M	[F _(7,95)_ = 17.300, *p* ≤ 0.001]	[F _(7,95)_ = 18.887, *p* ≤ 0.001]	[F _(7,95)_ = 33.352, *p* ≤ 0.001]

RT = resting time; DT = distance traveled; ST = stereotypic time; AT = ambulatory time; HC = horizontal counts; CZE = center zone entries; CZT = center zone time.

**Table 4 brainsci-13-01360-t004:** Locomotor activity assessed by OFT represented as mean values along with level of significance (two-way ANOVA).

	Age	Control vs. LP	Control vs. Control+LPS	Control vs. Control+DLT	Control vs. Control+DLT+LPS	LP vs. LP+LPS	LP vs. LP+DLT	LP vs. LP+DLT+LPS	Control+LPS vs. LP+LPS	Control+DLT vs. LP+DLT	Control+DLT+LPS vs. LP+DLT+LPS
RT (s)	1 M	[F _(1,95)_ = 38.631, *p* ≤ 0.001]	[F _(1,95)_ = 11.047, *p* ≤ 0.001]	[F _(1,95)_ = 7.105, *p* ≤ 0.015]	[F _(1,95)_ = 25.412, *p* ≤ 0.001]	[F _(1,95)_ = 15.910, *p* ≤ 0.001]	[F _(1,95)_ = 30.441, *p* ≤ 0.001]	[F _(1,95)_ = 66.914, *p* ≤ 0.001]	[F _(1,95)_ = 17.878, *p* ≤ 0.001]	[F _(1,95)_ = 36.098, *p* ≤ 0.001]	[F _(1,95)_ = 56.166, *p* ≤ 0.001]
	3 M	[F _(1,95)_ = 29.858, *p* ≤ 0.001]	-	-	[F _(1,95)_ = 15.993, *p* ≤ 0.001]	-	-	[F _(1,95)_ = 3.441, *p* ≤ 0.001]	[F _(1,95)_ = 13.893, *p* ≤ 0.001]	[F _(1,95)_ = 19.687, *p* ≤ 0.001]	[F _(1,95)_ = 17.306, *p* ≤ 0.001]
	6 M	[F _(1,95)_ = 20.701, *p* ≤ 0.001]	[F _(1,95)_ = 8.822, *p* ≤ 0.001]	[F _(1,95)_ = 24.253, *p* ≤ 0.001]	[F _(1,95)_ = 29.594, *p* ≤ 0.001]	[F _(1,95)_ = 2.988, *p* = 0.008]	[F _(1,95)_ = 9.914, *p* ≤ 0.001]	[F _(1,95)_ = 11.844, *p* ≤ 0.001]	[F _(1,95)_ = 14.867, *p* ≤ 0.001]	[F _(1,95)_ = 6.363, *p* ≤ 0.001]	[F _(1,95)_ = 2.952, *p* = 0.004]
DT (cm)	1 M	[F _(1,95)_ = 2.914, *p* = 0.005]	[F _(1,95)_ = 19.061, *p* ≤ 0.001]	[F _(1,95)_ = 36.054, *p* ≤ 0.001]	[F _(1,95)_ = 53.401, *p* ≤ 0.001]	[F _(1,95)_ = 33.401, *p* ≤ 0.001]	[F _(1,95)_ = 43.819, *p* ≤ 0.001]	[F _(1,95)_ = 58.240, *p* ≤ 0.001]	[F _(1,95)_ = 11.427, *p* ≤ 0.001]	[F _(1,95)_ = 4.851, *p* ≤ 0.001]	[F _(1,95)_ = 2.084, *p* = 0.041]
	3 M	[F _(1,95)_ = 28.132, *p* ≤ 0.001]	[F _(1,95)_ = 13.004, *p* ≤ 0.001]	[F _(1,95)_ = 34.270, *p* ≤ 0.001]	[F _(1,95)_ = 55.114, *p* ≤ 0.001]	[F _(1,95)_ = 11.461, *p* ≤ 0.001]	[F _(1,95)_ = 13.356, *p* ≤ 0.001]	[F _(1,95)_ = 51.358, *p* ≤ 0.001]	[F _(1,95)_ = 26.589, *p* ≤ 0.001]	F _(1,95)_ = 7.218, *p* ≤ 0.001]	[F _(1,95)_ = 24.376, *p* ≤ 0.001]
	6 M	[F _(1,95)_ = 48.928, *p* ≤ 0.001]	[F _(1,95)_ = 45.814, *p* ≤ 0.001]	[F _(1,95)_ = 56.384, *p* ≤ 0.001]	[F _(1,95)_ = 62.304, *p* ≤ 0.001]	[F _(1,95)_ = 8.798, *p* ≤ 0.001]	[F _(1,95)_ = 11.625, *p* ≤ 0.001]	[F _(1,95)_ = 23.680, *p* ≤ 0.001]	[F _(1,95)_ = 11.912, *p* ≤ 0.001]	[F _(1,95)_ = 4.169, *p* ≤ 0.001]	[F _(1,95)_ = 10.304, *p* ≤ 0.001]
ST (s)	1 M	[F _(1,95)_ = 18.233, *p* ≤ 0.001]	[F _(1,95)_ = 39.610, *p* ≤ 0.001]	[F _(1,95)_ = 73.058, *p* ≤ 0.001]	[F _(1,95)_ = 110.216, *p* ≤ 0.001]	[F _(1,95)_ = 33.039, *p* ≤ 0.001]	[F _(1,95)_ = 86.136, *p* ≤ 0.001]	[F _(1,95)_ = 99.779, *p* ≤ 0.001]	[F _(1,95)_ = 9.973, *p* ≤ 0.001]	[F _(1,95)_ = 33.785, *p* ≤ 0.001]	[F _(1,95)_ = 8.480, *p* ≤ 0.001]
	3 M	[F _(1,95)_ = 13.590, *p* ≤ 0.001]	[F _(1,95)_ = 13.829, *p* ≤ 0.001]	[F _(1,95)_ = 15.065, *p* ≤ 0.001]	[F _(1,95)_ = 19.783, *p* ≤ 0.001]	[F _(1,95)_ = 9.621, *p* ≤ 0.001]	[F _(1,95)_ = 10.179, *p* ≤ 0.001]	[F _(1,95)_ = 16.529, *p* ≤ 0.001]	[F _(1,95)_ = 9.381, *p* ≤ 0.001]	[F _(1,95)_ = 8.704, *p* ≤ 0.001]	[F _(1,95)_ = 10.386, *p* ≤ 0.001]
	6 M	[F _(1,95)_ = 11.466, *p* ≤ 0.001]	[F _(1,95)_ = 28.519, *p* ≤ 0.001]	[F _(1,95)_ = 16.824, *p* ≤ 0.001]	[F _(1,95)_ = 44.242, *p* ≤ 0.001]	[F _(1,95)_ = 24.074, *p* ≤ 0.001]	[F _(1,95)_ = 26.524, *p* ≤ 0.001]	[F _(1,95)_ = 38.302, *p* ≤ 0.001]	[F _(1,95)_ = 7.021, *p* ≤ 0.001]	[F _(1,95)_ = 21.166, *p* ≤ 0.001]	[F _(1,95)_ = 5.525, *p* ≤ 0.001]
AT (s)	1 M	[F _(1,95)_ = 6.847, *p* ≤ 0.001]	[F _(1,95)_ = 8.594, *p* ≤ 0.001]	[F _(1,95)_ = 16.244, *p* ≤ 0.001]	[F _(1,95)_ = 25.573, *p* ≤ 0.001]	[F _(1,95)_ = 4.997, *p* ≤ 0.001]	[F _(1,95)_ = 13.834, *p* ≤ 0.001]	[F _(1,95)_ = 22.498, *p* ≤ 0.001]	[F _(1,95)_ = 3.284, *p* = 0.003]	[F _(1,95)_ = 5.031, *p* ≤ 0.001]	[F _(1,95)_ = 3.773, *p* ≤ 0.001]
	3 M	[F _(1,95)_ = 4.537, *p* ≤ 0.001]	[F _(1,95)_ = 5.849, *p* ≤ 0.001]	[F _(1,95)_ = 7.443, *p* ≤ 0.001]	[F _(1,95)_ = 14.849, *p* ≤ 0.001]	[F _(1,95)_ = 4.717, *p* ≤ 0.001]	[F _(1,95)_ = 6.318, *p* ≤ 0.001]	[F _(1,95)_ = 23.060, *p* ≤ 0.001]	[F _(1,95)_ = 2.121, *p* = 0.037]	[F _(1,95)_ = 3.412, *p* = 0.001]	[F _(1,95)_ = 12.749, *p* ≤ 0.001]
	6 M	[F _(1,95)_ = 11.028, *p* ≤ 0.001]	[F _(1,95)_ = 14.201, *p* ≤ 0.001]	[F _(1,95)_ = 11.424, *p* ≤ 0.001]	[F _(1,95)_ = 21.262, *p* ≤ 0.001]	[F _(1,95)_ = 6.387, *p* ≤ 0.001]	[F _(1,95)_ = 6.545, *p* ≤ 0.001]	[F _(1,95)_ = 16.502, *p* ≤ 0.001]	[F _(1,95)_ = 3.213, *p* = 0.002]	[F _(1,95)_ = 6.149, *p* ≤ 0.001]	[F _(1,95)_ = 6.268, *p* ≤ 0.001]
HC	1 M	[F _(1,95)_ = 23.069, *p* ≤ 0.001]	[F _(1,95)_ = 20.677, *p* ≤ 0.001]	[F _(1,95)_ = 32.187, *p* ≤ 0.001]	[F _(1,95)_ = 41.210, *p* ≤ 0.001]	[F _(1,95)_ = 43.336, *p* ≤ 0.001]	[F _(1,95)_ = 35.695, *p* ≤ 0.001]	[F _(7195)_ = 55.135, *p* ≤ 0.001]	[F _(1,95)_ = 47.650, *p* ≤ 0.001]	[F _(1,95)_ = 28.354, *p* ≤ 0.001]	[F _(1,95)_ = 38.771, *p* ≤ 0.001]
	3 M	[F _(1,95)_ = 24.846, *p* ≤ 0.001]	[F _(1,95)_ = 19.705, *p* ≤ 0.001]	[F _(1,95)_ = 26.677, *p* ≤ 0.001]	[F _(1,95)_ = 32.427, *p* ≤ 0.001]	[F _(1,95)_ = 4.125, *p* ≤ 0.001]	[F _(1,95)_ = 8.358, *p* ≤ 0.001]	[F _(1,95)_ = 30.306, *p* ≤ 0.001]	[F _(1,95)_ = 7.338, *p* ≤ 0.001]	[F _(1,95)_ = 4.600, *p* ≤ 0.001]	[F _(1,95)_ = 20.797, *p* ≤ 0.001]
	6 M	[F _(1,95)_ = 37.229, *p* ≤ 0.001]	[F _(1,95)_ 60.865, *p* ≤ 0.001]	[F _(1,95)_ = 75.750, *p* ≤ 0.001]	[F _(1,95)_ = 85.633, *p* ≤ 0.001]	[F _(1,95)_ = 35.213, *p* ≤ 0.001]	[F _(1,95)_ = 39.288, *p* ≤ 0.001]	[F _(1,95)_ = 53.707, *p* ≤ 0.001]	[F _(1,95)_ = 11.577, *p* ≤ 0.001]	-	[F _(1,95)_ = 3.896, *p* ≤ 0.001]
CZE	1 M	[F _(1,95)_ = 8.089, *p* ≤ 0.001]	[F _(1,95)_ = 16.928, *p* ≤ 0.001]	[F _(1,95)_ = 18.679, *p* ≤ 0.001]	[F _(1,95)_ = 30.187, *p* ≤ 0.001]	[F _(1,95)_ = 11.424, *p* ≤ 0.001]	[F _(1,95)_ = 15.594, *p* ≤ 0.001]	[F _(1,95)_ = 25.350, *p* ≤ 0.001]	[F _(1,95)_ = 2.585, *p* = 0.012]	[F _(1,95)_ = 5.003, *p* ≤ 0.001]	[F _(1,95)_ = 3.252, *p* = 0.002]
	3 M	[F _(1,95)_ = 9.377, *p* ≤ 0.001]	[F _(1,95)_ = 19.025, *p* ≤ 0.001]	[F _(1,95)_ = 33.903, *p* ≤ 0.001]	[F _(1,95)_ = 49.051, *p* ≤ 0.001]	[F _(1,95)_ = 14.517, *p* ≤ 0.001]	[F _(1,95)_ = 32.370, *p* ≤ 0.001]	[F _(1,95)_ = 43.370, *p* ≤ 0.001]	[F _(1,95)_ = 4.869, *p* ≤ 0.001]	[F _(1,95)_ = 7.845, *p* ≤ 0.001]	[F _(1,95)_ = 3.697, *p* ≤ 0.001]
	6 M	[F _(1,95)_ = 7.881, *p* ≤ 0.001]	[F _(1,95) =_ 15.761, *p* ≤ 0.001]	[F _(1,95)_ = 24.955, *p* ≤ 0.001]	[F _(1,95)_ = 41.025, *p* ≤ 0.001]	[F _(1,95)_ = 14.602, *p* ≤ 0.001]	[F _(1,95)_ = 27.968, *p* ≤ 0.001]	[F _(1,95)_ = 42.339, *p* ≤ 0.001]	[F _(1,95)_ = 6.722, *p* ≤ 0.001]	[F _(1,95)_ = 10.894, *p* ≤ 0.001]	[F _(1,95)_ = 9.194, *p* ≤ 0.001]
CZT (s)	1 M	[F _(1,95)_ = 9.893, *p* ≤ 0.001]	[F _(1,95)_ = 15.937, *p* ≤ 0.001]	[F _(1,95)_ = 18.531, *p* ≤ 0.001]	[F _(1,95)_ = 24.720, *p* ≤ 0.001]	[F _(1,95)_ = 8.638, *p* ≤ 0.001]	[F _(1,95)_ = 5.839, *p* ≤ 0.001]	[F _(1,95)_ = 14.824, *p* ≤ 0.001]	-	[F _(1,95)_ = 2.572, *p* = 0.048]	[F _(1,95)_ = 3.606, *p* = 0.005]
	3 M	[F _(1,95)_ = 10.855, *p* ≤ 0.001]	[F _(1,95)_ = 14.778, *p* ≤ 0.001]	[F _(1,95)_ = 17.114, *p* ≤ 0.001]	[F _(1,95)_ = 26.1031, *p* ≤ 0.001]	[F _(1,95)_ = 7.753, *p* ≤ 0.001]	[F _(1,95)_ = 6.259, *p* ≤ 0.001]	[F _(1,95)_ = 15.248, *p* ≤ 0.001]	[F _(1,95)_ = 3.830, *p* = 0.002]	-	[F _(1,95)_ = 3.858, *p* = 0.002]
	6 M	[F _(1,95)_ = 11.701, *p* ≤ 0.001]	[F _(1,95) =_ 14.194, *p* ≤ 0.001]	[F _(1,95)_ = 16.460, *p* ≤ 0.001]	[F _(1,95)_ = 33.352, *p* ≤ 0.001]	[F _(1,95)_ = 5.599, *p* ≤ 0.001]	[F _(1,95)_ = 7.186, *p* ≤ 0.001]	[F _(1,95)_ = 21.651, *p* ≤ 0.001]	[F _(1,95)_ = 2.706, *p* = 0.042]	-	[F _(1,95)_ = 8.679, *p* ≤ 0.001]

**Table 5 brainsci-13-01360-t005:** Elevated plus maze data represented as mean values of % open arm entries (% OAE), % open arm time (% OAT), center zone entries (CZE) and center zone time (CZT) along with level of significance (one-way ANOVA).

	Age	Control vs. LP+LPS	Control vs. LP+DLT	Control vs. LP+LPS+DLT
% OAE	1 M	[F _(7,95)_ = 8.049, *p* ≤ 0.001]	[F _(7,95)_ = 12.819, *p* ≤ 0.001]	[F _(7,95)_ = 19.974, *p* ≤ 0.001]
	3 M	[F _(7,95)_ = 23.156, *p* ≤ 0.001]	[F _(7,95)_ = 16.259, *p* ≤ 0.001]	[F _(7,95)_ = 23.156, *p* ≤ 0.001]
	6 M	[F _(7,95)_ = 9.237, *p* ≤ 0.001]	[F _(7,95)_ = 14.676, *p* ≤ 0.001]	[F _(7,95)_ = 23.732, *p* ≤ 0.001]
% OAT	1 M	[F _(7,95)_ = 20.987, *p* ≤ 0.001]	[F _(7,95)_ = 29.663, *p* ≤ 0.001]	[F _(7,95)_ = 41.150, *p* ≤ 0.001]
	3 M	[F _(7,95) =_ 30.792, *p* ≤ 0.001]	[F _(7,95)_ = 103.914, *p* ≤ 0.001]	[F _(7,95)_ = 126.484, *p* ≤ 0.001]
	6 M	[F _(7,95)_ = 12.004, *p* ≤ 0.001]	[F _(7,95)_ = 12.755, *p* ≤ 0.001]	[F _(7,95)_ = 18.094, *p* ≤ 0.001]
CZE	1 M	[F _(7,95)_ = 8.948, *p* ≤ 0.001]	[F _(7,95)_ = 11.968, *p* ≤ 0.001]	[F _(7,95)_ = 19.565, *p* ≤ 0.001]
	3 M	[F _(7,95)_ = 8.635, *p* ≤ 0.001]	[F _(7,95)_ = 17.375, *p* ≤ 0.001]	[F _(7,95)_ = 26.115, *p* ≤ 0.001]
	6 M	[F _(7,95)_ = 21.089, *p* ≤ 0.001]	[F _(7,95)_ = 27.725, *p* ≤ 0.001]	[F _(7,95)_ = 13.933, *p* ≤ 0.001]
CZT (s)	1 M	[F _(7,95)_ = 13.816, *p* ≤ 0.001]	[F _(7,95)_ = 13.262, *p* ≤ 0.001]	[F _(7,95)_ = 18.227, *p* ≤ 0.001]
	3 M	[F _(7,95)_ = 16.950, *p* ≤ 0.001]	[F _(7,95)_ = 18.527, *p* ≤ 0.001]	[F _(7,95)_ = 27.690, *p* ≤ 0.001]
	6 M	[F _(7,95)_ = 25.785, *p* ≤ 0.001]	[F _(7,95)_ = 28.142, *p* ≤ 0.001]	[F _(7,95)_ = 43.407, *p* ≤ 0.001]

**Table 6 brainsci-13-01360-t006:** Elevated plus maze data represented as mean values of % open arm entries (% OAE), % open arm time (% OAT), center zone entries (CZE) and center zone time (CZT) along with level of significance (two-way ANOVA).

	Age	Control vs. LP	Control vs. Control+LPS	Control vs. Control+DLT	Control vs. Control+DLT+LPS	LP vs. LP+LPS	LP vs. LP+DLT	LP vs. LP+DLT+LPS	Control+LPS vs. LP+LPS	Control+DLT vs. LP+DLT	Control+DLT+LPS vs. LP+DLT+LPS
% OAE	1 M	[F _(1,95)_ = 4.562, *p* ≤ 0.001]	[F _(1,95)_ = 4.339, *p* ≤ 0.001]	[F _(1,95)_ = 10.084, *p* ≤ 0.001]	[F _(1,95)_ = 15.952, *p* ≤ 0.001]	[F _(1,95)_ = 3.487, *p* = 0.010]	[F _(1,95)_ = 8.257, *p* ≤ 0.001]	[F _(1,95)_ = 15.412, *p* ≤ 0.001]	[F _(1,95)_ = 3.710, *p* = 0.009]	[F _(1,95)_ = 2.735, *p* = 0.017]	[F _(1,95)_ = 4.022, *p* = 0.001]
	3 M	[F _(1,95)_ = 6.927, *p* ≤ 0.001]	[F _(1,95)_ = 4.553, *p* ≤ 0.001]	[F _(1,95)_ = 10.596, *p* ≤ 0.001]	[F _(1,95)_ = 14.716, *p* ≤ 0.001]	[F _(1,95)_ = 11.578, P ≤ 0.001]	[F _(1,95)_ = 9.332, *p* ≤ 0.001]	[F _(1,95)_ = 16.229, *p* ≤ 0.001]	[F _(1,95)_ = 13.952, *p* ≤ 0.001]	[F _(1,95)_ = 5.664, *p* ≤ 0.001]	[F _(1,95)_ = 8.441, *p* ≤ 0.001]
	6 M	[F _(1,95)_ = 5.431, *p* = 0.001]	[F _(1,95)_ = 4.894, *p* ≤ 0.001]	[F _(1,95)_ = 9.152, *p* ≤ 0.001]	[F _(1,95)_ = 23.732, *p* ≤ 0.001]	[F _(1,95)_ = 3.806, *p* = 0.001]	[F _1,95)_ = 9.245, *p* ≤ 0.001]	[F _(1,95)_ = 18.301, *p* ≤ 0.001]	[F _(1,95)_ = 4.344, *p* ≤ 0.001]	[F _(1,95)_ = 5.524, *p* = ≤ 0.001]	[F _(1,95)_ = 7.197, *p* ≤ 0.001]
% OAT	1 M	[F _(1,95)_ = 3.573, *p* = 0.004]	-	[F _(1,95)_ = 8.545, *p* ≤ 0.001]	[F _(1,95)_ = 20.334, *p* ≤ 0.001]	[F _(1,95)_ = 17.414, *p* ≤ 0.001]	[F _(1,95)_ = 26.091, *p* ≤ 0.001]	[F _(1,95)_ = 37.577, *p* ≤ 0.001]	[F _(1,95)_ = 18.738, *p* ≤ 0.001]	[F _(1,95)_ = 21.118, *p* ≤ 0.001]	[F _(1,95)_ = 20.816, *p* ≤ 0.001]
	3 M	[F _(1,95)_ = 14.714, *p* ≤ 0.001]	[F _(1,95)_ = 14.147, *p* ≤ 0.001]	[F _(1,95)_ = 101.005, *p* ≤ 0.001]	[F _(1,95)_ = 108.576, *p* ≤ 0.001]	[F _(1,95)_ = 16.506, *p* ≤ 0.001]	[F _(1,95)_ = 89.628, *p* ≤ 0.001]	[F _(1,95)_ = 112.198, *p* ≤ 0.001]	[F _(1,95)_ = 16.644, *p* ≤ 0.001]	-	[F _(1,95)_ = 17.908, *p* ≤ 0.001]
	6 M	[F _(1,95)_ = 5.454, *p* ≤ 0.001]	[F _(1,95)_ = 7.897, *p* ≤ 0.001]	[F _(1,95)_ = 8.960, *p* ≤ 0.001]	[F _(1,95)_ = 13.212, *p* ≤ 0.001]	[F _(1,95)_ = 6.550, *p* = *p* ≤ 0.001]	[F _(1,95)_ = 7.301, *p* ≤ 0.001]	[F _(1,95)_ = 12.650, *p* ≤ 0.001]	[F _(1,95)_ = 4.107, *p* ≤ 0.001]	[F _(1,95)_ = 3.795, *p* = 0.002]	[F _(1,95)_ = 4.883, *p* ≤ 0.001]
CZE	1 M	[F _(1,95)_ = 4.059, *p* = 0.001]	[F _(1,95)_ = 5.828, *p* ≤ 0.001]	[F _(1,95)_ = 5.412, *p* ≤ 0.001]	[F _(1,95)_ = 16.339, *p* ≤ 0.001]	[F _(1,95)_ = 4.889, *p* ≤ 0.001]	[F _(1,95)_ = 7.909, *p* ≤ 0.001]	[F _(1,95)_ = 15.507, *p* ≤ 0.001]	[F _(1,95)_ = 3.120, *p* = 0.013]	[F _(1,95)_ = 6.556, *p* ≤ 0.001]	[F _(1,95)_ = 3.226, *p* = 0.011]
	3 M	[F _(1,95)_ = 5.477, *p* ≤ 0.001]	-	[F _(1,95)_ = 7.793, *p* ≤ 0.001]	[F _(1,95)_ = 22.744, *p* ≤ 0.001]	[F _(1,95)_ = 3.158, *p* = 0.009]	[F _(1,95)_ = 11.898, *p* ≤ 0.001]	[F _(1,95)_ = 20.638, *p* ≤ 0.001]	[F _(1,95)_ = 6.740, *p* ≤ 0.001]	[F _(1,95)_ = 9.582, *p* ≤ 0.001]	[F _(1,95)_ = 3.371, *p* = 0.006]
	6 M	[F _(1,95)_ = 7.635, *p* ≤ 0.001]	[F _(1,95)_ = 17.362, *p* ≤ 0.001]	[F _(1,95)_ = 25.452, *p* ≤ 0.001]	[F _(1,95)_ = 32.543, *p* ≤ 0.001]	[F _(1,95)_ = 13.453, *p* ≤ 0.001]	[F _(1,95)_ = 20.089, *p* ≤ 0.001]	[F _(1,95)_ = 30.361, *p* ≤ 0.001]	[F _(1,95)_ = 3.727, *p* ≤ 0.001]	[F _(1,95)_ = 2.273, *p* = 0.026]	[F _(1,95)_ = 5.454, *p* ≤ 0.001]
CZT (s)	1 M	[F _(1,95)_ = 5.534, *p* ≤ 0.001]	[F _(1,95)_ = 10.791, *p* ≤ 0.001]	[F _(1,95)_ = 13.037, *p* ≤ 0.001]	[F _(1,95)_ = 14.542, *p* ≤ 0.001]	[F _(1,95)_ = 8.282, *p* ≤ 0.001]	[F _(1,95)_ = 7.728, *p* ≤ 0.001]	[F _(1,95)_ = 12.693, *p* ≤ 0.001]	[F _(1,95)_ = 3.025, *p* = 0.031]	-	[F _(1,95) =_ 3.684, *p* = 0.004]
	3 M	[F _(1,95)_ = 7.918, *p* ≤ 0.001]	[F _(1,95)_ = 10.902, *p* ≤ 0.001]	[F _(1,95)_ = 14.933, *p* ≤ 0.001]	[F _(1,95)_ = 21.890, *p* ≤ 0.001]	[F _(1,95)_ = 9.032, *p* ≤ 0.001]	[F _(1,95)_ = 10.609, *p* ≤ 0.001]	[F _(1,95)_ = 19.772, *p* ≤ 0.001]	[F _(1,95)_ = 6.048, *p* ≤ 0.001]	[F _(1,95)_ = 3.594, *p* = 0.003]	[F _(1,95)_ = 5.800, *p* ≤ 0.001]
	6 M	[F _(1,95)_ = 15.265, *p* ≤ 0.001]	[F _(1,95)_ = 23.219, *p* ≤ 0.001]	[F _(1,95)_ = 20.098, *p* ≤ 0.001]	[F _(1,95)_ = 35.273, *p* ≤ 0.001]	[F _(1,95)_ = 10.521, *p* ≤ 0.001]	[F _(1,95)_ = 12.878, *p* ≤ 0.001]	[F _(1,95)_ = 28.142, *p* ≤ 0.001]	[F _(1,95)_ = 2.566, *p* = 0.025]	[F _(1,95)_ = 8.044, *p* ≤ 0.001]	[F _(1,95)_ = 8.134, *p* ≤ 0.001]

**Table 7 brainsci-13-01360-t007:** Light and dark box data represented as mean values of % light zone entries (% LZE), % light zone time (% LZT), along with level of significance (one-way ANOVA).

	Age	Control vs. LP+LPS	Control vs. LP+DLT	Control vs. LP+LPS+DLT
% LZE	1 M	[F _(7,95)_ = 11.996, *p* ≤ 0.001]	[F _(7,95)_ = 12.475, *p* ≤ 0.001]	[F _(7,95)_ = 15.620, *p* ≤ 0.001]
	3 M	[F _(7,95)_ = 9.523, *p* ≤ 0.001]	[F _(7,95) =_ 10.163, *p* ≤ 0.001]	[F _(7,95)_ = 16.124, *p* ≤ 0.001]
	6 M	[F _(7,95)_ = 11.723, *p* ≤ 0.001]	[F _(7,95)_ = 12.827, *p* ≤ 0.001]	[F _(7,95)_ = 19.761, *p* ≤ 0.001]
% LZT	1 M	[F _(7,95)_ = 8.413, *p* ≤ 0.001]	[F _(7,95)_ = 9.659, *p* ≤ 0.001]	[F _(7,95)_ = 9.511, *p* ≤ 0.001]
	3 M	[F _(7,95)_ = 9.121, *p* ≤ 0.001]	[F _(7,95)_ = 10.434, *p* ≤ 0.001]	[F _(7,95)_ = 13.099, *p* ≤ 0.001]
	6 M	[F _(7,95)_ = 13.811, *p* ≤ 0.001]	[F _(7,95)_ = 20.132, *p* ≤ 0.001]	[F _(7,95)_ = 67.694, *p* ≤ 0.001]

**Table 8 brainsci-13-01360-t008:** Light and dark box data represented as mean values of % light zone entries (% LZE), % light zone time (% LZT), along with level of significance (two-way ANOVA).

	Age	Control vs. LP	Control vs. Control+LPS	Control vs. Control+DLT	Control vs. Control+DLT+LPS	LP vs. LP+LPS	LP vs. LP+DLT	LP vs. LP+DLT+LPS	Control+LPS vs. LP+LPS	Control+DLT vs. LP+DLT	Control+DLT+LPS vs. LP+DLT+LPS
% LZE	1 M	[F _(1,95)_ = 4.020, *p* = 0.001]	[F _(1,95)_ = 4.722, *p* ≤ 0.001]	[F _(1,95)_ = 7.770, *p* ≤ 0.001]	[F _(1,95)_ = 11.052, *p* ≤ 0.001]	[F _(1,95)_ = 7.976, *p* ≤ 0.001]	[F _(1,95)_ = 8.455, *p* ≤ 0.001]	[F _(1,95) =_ 11.599, *p* ≤ 0.001]	[F _(1,95)_ = 7.274, *p* ≤ 0.001]	[F _(1,95)_ = 4.705, *p* ≤ 0.001]	[F _(1,95)_ = 4.567, *p* ≤ 0.001]
	3 M	[F _(1,95)_ = 5.345, *p* ≤ 0.001]	[F _(1,95)_ = 3.643, *p* = 0.004]	[F _(1,95)_ = 4.558, *p* ≤ 0.001]	[F _(1,95)_ = 11.094, *p* ≤ 0.001]	[F _(1,95)_ = 4.178, *p* ≤ 0.001]	[F _(1,95)_ = 4.817, *p* ≤ 0.001]	[F _(1,95)_ = 10.779, *p* ≤ 0.001]	[F _(1,95)_ = 5.880, *p* ≤ 0.001]	[F _(1,95)_ = 5.605, *p* ≤ 0.001]	[F _(1,95)_ = 5.030, *p* ≤ 0.001]
	6 M	[F _(1,95)_ = 6.172, *p* ≤ 0.001]	[F _(1,95)_ = 7.316, *p* ≤ 0.001]	[F _(1,95)_ = 8.367, *p* ≤ 0.001]	[F _(1,95)_ = 14.752, *p* ≤ 0.001]	[F _(1,95)_ = 5.551, *p* ≤ 0.001]	[F _(1,95)_ = 6.655, *p* ≤ 0.001]	[F _(1,95)_ = 13.619, *p* ≤ 0.001]	[F _(1,95)_ = 4.408, *p* ≤ 0.001]	[F _(1,95)_ = 4.460, *p* ≤ 0.001]	[F _(1,95)_ = 5.039, *p* ≤ 0.001]
% LZT	1 M	[F _(1,95)_ = 4.856, *p* ≤ 0.001]	[F _(1,95)_ = 4.287, *p* ≤ 0.001]	[F _(1,95)_ = 5.543, *p* ≤ 0.001]	[F _(1,95)_ = 9.548, *p* ≤ 0.001]	[F _(1,95)_ = 3.557, *p* ≤ 0.001]	[F _(1,95)_ = 4.265, *p* ≤ 0.001]	[F _(1,95)_ = 8.956, *p* ≤ 0.001]	[F _(1,95)_ = 4.124, *p* ≤ 0.001]	[F _(1,95)_ = 3.578, *p* ≤ 0.001]	[F _(1,95)_ = 4.265, *p* ≤ 0.001]
	3 M	[F _(1,95)_ = 4.480, *p* ≤ 0.001]	[F _(1,95)_ = 5.040, *p* ≤ 0.001]	[F _(1,95)_ = 6.210, *p* ≤ 0.001]	[F _(1,95)_ = 12.388, *p* ≤ 0.001]	[F _(1,95)_ = 5.180, *p* ≤ 0.001]	[F _(1,95)_ = 5.954, *p* ≤ 0.001]	[F _(1,95)_ = 15.652, *p* ≤ 0.001]	[F _(1,95)_ = 4.619, *p* ≤ 0.001]	[F _(1,95)_ = 4.224, *p* ≤ 0.001]	[F _(1,95)_ = 7.744, *p* ≤ 0.001]
	6 M	[F _(1,95)_ = 5.798, *p* ≤ 0.001]	[F _(1,95)_ = 6.054, *p* ≤ 0.001]	[F _(1,95)_ = 6.490, *p* ≤ 0.001]	[F _(1,95)_ = 17.577, *p* ≤ 0.001]	[F _(1,95)_ = 3.713, *p* ≤ 0.001]	[F _(1,95)_ = 7.301, *p* ≤ 0.001]	[F _(1,95)_ = 16.261, *p* ≤ 0.001]	[F _(1,95)_ = 3.457, *p* ≤ 0.001]	[F _(1,95)_ = 6.607, *p* ≤ 0.001]	[F _(1,95)_ = 4.482, *p* ≤ 0.001]

**Table 9 brainsci-13-01360-t009:** Retention time (RT) assessed in rotarod test represented as mean (s) along with the level of significance (one-way ANOVA).

	Age	Control vs. LP+LPS	Control vs. LP+DLT	Control vs. LP+LPS+DLT
RT (s)	3 M	[F _(7,95)_ = 8.476, *p* ≤ 0.001]	[F _(7,95)_ = 10.659, *p* ≤ 0.001]	[F _(7,95)_ = 13.238, *p* ≤ 0.001]
	6 M	[F _(7,95)_ = 14.982, *p* ≤ 0.001]	[F _(7,95)_ = 14.755, *p* ≤ 0.001]	[F _(7,95)_ = 28.235, *p* ≤ 0.001]

**Table 10 brainsci-13-01360-t010:** Retention time (RT) assessed in rotarod test represented as mean (s) along with the level of significance (two-way ANOVA).

	Age	Control vs. LP	Control vs. Control+LPS	Control vs. Control+DLT	Control vs. Control+DLT+LPS	LP vs. LP+LPS	LP vs. LP+DLT	LP vs. LP+DLT+LPS	Control+LPS vs. LP+LPS	Control+DLT vs. LP+DLT	Control+DLT+LPS vs. LP+DLT+LPS
RT (s)	3 M	[F _(1,95)_ = 3.542, *p* ≤ 0.001]	[F _(1,95)_ = 4.801, *p* ≤ 0.001]	[F _(1,95) =_ 5.486, *p* ≤ 0.001]	[F _(1,95)_ = 9.791, *p* ≤ 0.001]	[F _(1,95)_ = 4.447, *p* ≤ 0.001]	[F _(1,95)_ = 6.630, *p* = 0.001]	[F _(1,95)_ = 9.209, *p* ≤ 0.001]	[F _(1,95)_ = 3.175, *p* = 0.002]	[F _(1,95) =_ 4.666, *p* ≤ 0.001]	[F _(1,95)_ = 3.447, *p* = 0.001]
	6 M	[F _(1,95)_ = 4.431, *p* ≤ 0.001]	[F _(1,95)_ = 7.044, *p* ≤ 0.001]	[F _(1,95)_ = 10.955, *p* ≤ 0.001]	[F _(1,95)_ = 20.354, *p* ≤ 0.001]	[F _(1,95)_ = 11.616, *p* ≤ 0.001]	[F _(1,95) =_ 11.373, *p* ≤ 0.001]	[F _(1,95)_ = 25.811, *p* ≤ 0.001]	[F _(1,95))_ = 9.003, *p* ≤ 0.001]	[F _(1,95)_ = 4.849, *p* ≤ 0.001]	[F _(1,95)_ = 9.888, *p* ≤ 0.001]

## Data Availability

The data will be made available on request.

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
