# Peer review of "Multiple Early Life Stressors as Risk Factors for Neurodevelopmental Abnormalities in the F1 Wistar Rats"

_brainsci, 2023, doi:10.3390/brainsci13101360_

Round 1
Reviewer 1 Report
In this work, Mujtaba et al use a triple hit developmental rat model to ascertain the additive/synergistic effects of protein malnourishment, infection(LPS) and exposure to the toxicant pyrethroid on disease-relevant behavior later in life. They find that single of multiple exposures increase the prevalence of altered behavior, including hyperactivity, reduced anxiety. They do so in additive or synergistic manner. The studies are well performed and new. Although not very innovative conceptually, they add important value to the multiple-developmental-hits hypothesis in the etiology of neurological conditions.
I have the following specific comments:
It is not clear why the reduced protein diet of LP rats was maintained until adulthood, when the purpose was to assess the impact of a developmental hit. It probably should have been eliminated at weaning age. Thus the authors need to acknowledge this caveat in experimental design, as the LP insult is maintained throughout life and may impact many physiological systems in adulthood, not just critical brain developmental processes that occur during particular developmental windows.
Although there might not be differences between males and females, perhaps and ANOVA comparison taken sex into account could also be provided. This is highly relevant given the sexually dimorphic prevalence of many neuropsychiatric conditions.
The authors mention in several instances and sections that they measured “motor impulsivity”, refer to “impulsive behaviors” and conclude about “impulsiveness”. This is confusing, as “impulsivity” is a different concept: the speed to react to certain stimuli, which is not really measured by the tests performed. “Locomotor activity” should suffice. Also, grooming is more related to anxiety or compulsive behavior, not with impulsive behavior.
Lines 451-452. This statement is not correct. Increased time in the open arms is associated with low anxiety.
Lines 695-696. The authors state: “ As the heritability of ADHD and schizophrenia is approximately less than 1% [117]”.
This is not correct. Multiple studies have shown that the heritability of ADHD and schizophrenia is very high, close to 80%. In reference 117, it is actually stated that ADHD has high heritability. Also, reference 117 refers to ADHD only, not schizophrenia.
Lines 554-557. This is a bit of an overstatement, as the work is performed in rats. Rather, the authors provide evidence that a multiple developmental hit can triggers behavioral abnormalities in adulthood of relevance to human disease.
Line 484: The number of entries in the light/dark box is not really a measurement of locomotor activity, but rather anxiety. Time immobile or distance traveled are.
Lines 220-221, 255, 276: 9 am to 5.30 pm is a rather long interval of testing time to minimize circadian effects. As the same time interval used for the elevated plus maze?
Introduction and Discussion: Somewhat long and repetitive and redundant. Some concision and consolidation will facilitate the reading
Conclusion: ADHD, probably more deserving that schizophrenia, is not clearly mentioned. “Attention deficit symptoms” is probably not good wording. I would caution to something along the lines of “behavioral and locomotor characteristics of relevance to ADHD”.
Perhaps the significance tables could be presented in a more simplified or reduced-size manner, and the colors of the graphs bars improved for better discernment.
Minor Editing suggestions:
Line 35: “Adequate” is not the right word in this context.
Line 38, 98: “exposure to…” “not exposure of..”
Line 43: “due” could be deleted
Lines 56-58: Something is not grammatically right with this sentence
Line 84: “alleviating” is not the right word to use in this context.
Line 114: Please define PMN at first mention in the text
Line 135: For clarity, I would suggest: “…to achieve a state of protein malnutrition before conception.”
Line 705: There are no synergistic exposures. Exposures are additive. The effects of exposures are/could be synergistic.
There are a few instances in which the authors use wrong prepositions, unsuitable words or incorrect grammar, but they are generally minor and do give rise to confusion about meaning
Author Response
We have categorically answered all the queries and uploaded as word file.

Reviewer 2 Report
1. Introduction section: authors have already mentioned various maternal, pre and postnatal behavioral and cognitive disorders caused by DLT, protein malnutrition, and perinatal immune activation reported in both humans and animals. So, how this study is novel or any sort of addition to the existing information? Combining all the risk factors will aggravate the immunological response to every physiological extent. Isn’t that obvious? It would have been much appreciated if the authors have proposed any mechanisms underlying immunological responses and/or resultant behavioral alterations instead of combining all the stressors and assessing similar behavioral dysfunctions.
So, the major question here is the relevance of this study. This question should have already been considered by the senior authors (of this manuscript) before designing this experiment.
2. Methodology: There are numerous anomalies and redundancy in the methodology that raises the question about the rationality of this work. Mentioning here a few:
a. Why control females were fed with a 20% protein diet? How this diet is different from the diet provided to the male mating partners?
b. How the protein malnutrition animal model in this study was validated? Is 15 days of diet restriction enough to generate a protein-malnourished physiology? The authors must present some data to confirm the validity of this maternal PMN model.
d. Pg 3 line 133, “Fresh three month age virgin female rats………..’, What is this fresh mean? The authors have already mentioned that the females were virgins. Thus, avoid pleonasm.
c. Pg 3 line 142, “Litter size was adjusted to eight per dam……..”, Curious to know how authors have adjusted the litter size per dam.
d. Authors didn’t mention any signs of conceiving or maternity issues with protein-malnourished females. Even if the protein-malnourished females were equally fertile and healthy like a normal female mouse, which would be surprising, then the author must mention that too. As suggested before, PMN model validation is a huge lacuna in this whole work.
e. Author also didn’t mention any mortality data. It was well established that protein deficiency is related to pregnancy and embryonic complications. Though, this study didn’t mention any such issues.
f. Pg 5 line 186-189, No need to repeat the drug and dosage information again. Most of it is already mentioned in the group allocation. Just add the diluent information in that section.
g. Pg 5 line 192, “To counter any bias, Control animals………”, it should be ‘control’. Correct the capital word use.
h. Pups were injected without anaesthesia. Wondering under which animal ethical clearance authors have opted for this method. Even to euthanize an animal, anaesthesia is the first compulsory recommended step. How come pups in this study were never provided with any anaesthesia before injection? Such practice raises a serious ethical issue about the experiments conducted in this study. Intriguingly under what ethical and scientific considerations, the Institutional animal ethical committee have approved such practices? Explain.
i. Surprising, that after getting such severe treatment like PMN, DLT and LPS, there was no mortality reported.
j. section 2.2.1.: In the open field test, line 216-218, “animals spending more time in the centre zone was considered as indicative of low anxiety”, Is there any scientific explanation for considering the centre zone indicative of low anxiety?
3. Results: Graph colour of Control and Cont + treated group vs LP and LP + treated group are quite similar. Choose distinct colours for the graphs.
Fig 2B: In pictures, what is the rationale to show the comparison of the control animal with LP and LP+ treated (DLT, LPS) groups? Whilst authors have generated and assessed control + treatment (DLT, LPS) animals, so why the respective groups are not compared with each other?
Fig. 6. Picture quality is very poor. Increase the image resolution.
4. Discussion: The discussion is too long.
Line 552-554: The authors stated that “dual-hit hypothesis alone is insufficient to establish a relation between stressors and neurological diseases”, please explain on what ground they came to this conclusion.
Line 554-446: “……exposure to the early onset symptoms of neurological disease like schizophrenia and ADHD”. How come, low-anxiety, hyperactivity, and motor impulsivity cover the symptoms of schizophrenia and ADHD-like mental illness? Authors should pay more attention and read more literature while comparing such mental illnesses, like schizophrenia and ADHD (with a huge spectrum of cognitive symptoms) and claiming the similarity with 2-3 behavioural alterations of their own study.
Line 699-702: “Thus, it is suggested that cumulative early life exposure to maternal protein malnutrition, bacterial infection and pesticide exposure may act as crucial risk factors for schizophrenia, ADHD and other neuropsychiatric disorders and may pre-dispose individuals to develop such psychopathologies during adolescence and adulthood.”
As mentioned before authors have outreached to claim the similarity of the reported behavioral alterations with neuropsychiatric disorders like ADHD and schizophrenia. Correct it.
Minor editing of the English language is required.
Author Response

(The authors gave the same response as above.)

Round 2
Reviewer 1 Report
Thank you for the responsiveness.
Still, none of the test performed do measure impulsivity, but "impulsivity", "impulsiveness" and similar are still mentioned in several instances, including the key words of the article. Locomotor activity, physical activity, spontaneous activity or similar should be substituted.
Author Response
Thanks for your suggestions.
As suggested we have improved both the introduction and the conclusion parts.
The word impulsivity/impulsiveness has been substituted appropriately as suggested.
Now nowhere we are claiming the such stressors cause impulsivity.
Reviewer 2 Report
Introduction:
There is any evidence or statistics report from WHO, and/or national or international environmental or health agency, which shows that any developing country or countries are facing such incidence of PMN, environmental toxicant (DLT), and viral infection conditions altogether?
There are several reports of toxicants, PMN and/or viral infections exposure together (any two of these stressors) and/or singly causing all sort of behavioral and cognitive alterations (also mentioned in your introduction section). Is there any incidence report of having these three stressors exposure together, or it’s just an assumption in a hyperbolic manner? Authors not only lack the tangible objective but also inconsiderate and ambiguous in methodology and interpretation. Seems like they are really deficient in any knowledge about this work area.
2.Methodology:
a. Author should mention the diet plan in methodology also. And if the diet plan is procured from previous research, then mention that too. Again, if it is your lab established model then provide the reference or some biochemical, pathological and BMI index in supplementary information to confirm the model validation. In lack of this validation, the whole work authenticity is highly questionable.
b. Indeed. So, it wouldn’t be a big issue to provide the model validation data.
Also, the provided references are absolute mislead for the present work. Hard to understand why authors took references from the studies totally different from their own work. Marwarha et al., 2017 is about palmitate-enriched diet (protein 23.6% with palmitic acid 2.20%). Please explain how this work animal model is like your work model. Rushmore et al., 2021 animal model id 25% (cont) and 6% (LP) and they fed for 5 weeks. So, your work has nothing to do with these animal models. As I mentioned before, animal model validation is the weakest part of this whole study. Authors are warned to provide misleading information.
d. Please provide reference of any ‘usual practice followed in similar diet related experiment’. If the pups are of the same experimental group and in the end, you have combined them all for statistics, then what’s the rationale for separating them from their mother and foster feeding?
e. It's believable, as you have fed the females for 15 days of LP diet only. So, surely it was not sufficient enough to generate the protein deficient physiology, like the manuscript claimed and if the LP-fed females were really mimicking the PMN state, then include some model validation data. Otherwise, this whole study is an absolute mislead.
f & j. This claim is plausible as the model was not the precise mimic of PMN model. Otherwise, there is no way the authors haven’t experienced any animal mortality or pregnancy-related issues in the whole study tenure.
i. Surprise to read this ‘Scientific and ethical relevance’. Senior authors of this manuscript have been working on the development model for so long and they never heard of hypothermia-induced anesthesia. Just to brush up on the author’s knowledge, anesthesia is not only induced by anesthetic drugs. This group will benefit from some good reading about the techniques of developing postnatal animal models. Researchers have been using hypothermia for decades for postnatal ICV/IC injections on P3-5-day-old pups and even for severe toxicology studies more rigorous than the present one. Injecting a conscious 3 day old pup (making constant body movements) with Stoelting Nanoinjector and Hamilton micro-syringe with a gauge size of 22s, and also in quick speed, is not going to provide any comfort to them in anyway. Actually that’s extremely painful and damaging (physically) to them.
Authors are suggested to submit the approved part of animal ethical clearance where they have mentioned the method of pup injection.
4. Discussion:
In the sentence, “Due to the large spectrum of stressors and their variable modes of action, the dual-hit hypothesis is not sufficient to establish a clear relation between stressors and neurological diseases (30, 54, 55). While checking upon the references 54 and 55 are the studies advocating the dual-hit hypothesis and 30 is the self-citation. So, how come authors are using these references to support their claim that dual hit is not sufficient to establish the relation between stressors and neurological diseases? Authors are strictly suggested to read and understand the articles carefully before citing them. And your statement on ‘absence of genetic hit dual hypothesis become insufficient’ sounds nebulous cause you already said that genetic hit is not sufficient to cause the stress, so why the absence or presence of genetic hit is necessary to effectuate the state of dual hit? Apparently, authors are vague in their own explanation.
Line 651-656, explains the present study’s interpretation of anxiety-like behaviour in light/dark box. Why these lines are referred to explain the ‘onset of the symptoms not the disease’. Authors should read their own manuscript VERY carefully before answering the queries and providing the references.
Line 663-665, ‘thus environmental………. through epigenetic mechanisms’. Surprising how authors can even about speculate the epigenetic mechanism, when they performed none of the gene and/or cell-level assessment and it’s just one generation behaviour study. Health condition from, a PMN mother to pup and then stressors exposure; doesn’t this should be explained on metabolic alongwith toxicant related alterations and for the mechanism interpretation like epigenetic, authors should present some essential cell and/or gene level assessment.
Conclusion
Line 670-672: ‘The present results indicates…………………and low-anxiety phenotype at adulthood. Low anxiety and hyperactivity are two contradictory symptoms of any of the claimed disorders symptoms, i.e., ADHD and/or schizophrenia.
Again, authors need some serious reading and understanding about the neuropsychiatric disorders. It is clinically proven that, during ADHD or schizophrenia, symptoms are anxiety and hyperactivity. On psychological level, a person cannot present hyperactivity with low anxiety emotions neither in mentioned psychiatric disorders nor in any mental illness (reported so far).
Authors are again strictly suggested to read more and apprehend the information before claiming and/or writing such conclusions. Such erroneous claims and understanding reflects the elusiveness of the author’s scientific apprehension.
Moderate editing of English language required.
Author Response
We have uploaded a file responding to all the queries.
